# O$^2$-Searcher: A Searching-based Agent Model for Open-Domain Open-Ended Question Answering

**Jianbiao Mei**[1,2,*], **Tao Hu**[3,2,*], **Daocheng Fu**[2,4,*], **Licheng Wen**[2], **Xuemeng Yang**[2], **Rong Wu**[1,2],
**Pinlong Cai**[2], **Xinyu Cai**[2], **Xing Gao**[2], **Yu Yang**[1], **Chengjun Xie**[3], **Botian Shi**[2,†], **Yong Liu**[1,5,†],
**Yu Qiao**[2]

[1] *Zhejiang University* [2] *Shanghai Artificial Intelligence Laboratory* [3] *University of Science and Technology of China*
[4] *Fudan University* [5] *State Key Laboratory of Industrial Control Technology*

**Reviewed on OpenReview:** *https://openreview.net/forum?id=rbIKFKFEeU*

## Abstract

Large Language Models (LLMs), despite their advancements, are fundamentally limited by their static parametric knowledge, hindering performance on tasks requiring open-domain up-to-date information. While enabling LLMs to interact with external knowledge environments is a promising solution, current efforts primarily address closed-end problems. Open-ended questions, which are characterized by lacking a standard answer or providing non-unique and diverse answers, remain underexplored. To bridge this gap, we present O$^2$-Searcher, a novel search agent leveraging reinforcement learning to effectively tackle both open-ended and closed-ended questions in the open domain. O$^2$-Searcher leverages an efficient, locally simulated search environment for dynamic knowledge acquisition, effectively decoupling the external world knowledge from the model's sophisticated reasoning processes. It employs a unified training mechanism with meticulously designed reward functions, enabling the agent to identify problem types and adapt different answer generation strategies. Furthermore, to evaluate performance on complex open-ended tasks, we construct O$^2$-QA, a high-quality benchmark featuring 300 manually curated, multi-domain open-ended questions with associated web page caches. Extensive experiments show that O$^2$-Searcher, using only a 3B model, significantly surpasses leading LLM agents on O$^2$-QA. It also achieves SOTA results on various closed-ended QA benchmarks against similarly-sized models, while performing on par with much larger ones. Code is available at https://github.com/KnowledgeXLab/O2-Searcher.

## 1 Introduction

Large Language Models (LLMs) have demonstrated remarkable progress in diverse tasks such as mathematical reasoning Shao et al. (2024) and code generation Li et al. (2025a), owing to their advanced understanding and generative capabilities. Despite these advancements, the capability boundaries of current LLMs are fundamentally bounded by their parametric knowledge, the information encoded during pre-training, which serves as the basis for their responses Kim et al. (2025). This architecture faces inherent constraints: (1) Real-world knowledge is both dynamic and unbounded, making it impossible for any finite model to fully encapsulate its breadth and evolution. (2) Information becomes obsolete rapidly (e.g., news updates may invalidate prior facts overnight), novel disciplines and discoveries continually arise, and specialized domains often demand expertise far beyond the scope of general training data. As a result, dependence on this static "snapshot" knowledge inevitably hampers performance on open-domain tasks that require real-time updates, deep specialization, or cross-domain integration, frequently leading to factual inaccuracies or hallucinations Peng et al. (2023); Ye et al. (2023); Zheng et al. (2025a); Kim et al. (2025).

---

*Equal Contribution; †Corresponding Authors: yongliu@iipc.zju.edu.cn, shibotian@pjlab.org.cn

One way to address the above challenge is to let LLMs interact with external knowledge environments, decoupling the storage of external world knowledge from the model's core reasoning capabilities. The central idea of this strategy is to focus on training the model to master a set of capabilities for efficiently utilizing external knowledge resources. In other words, we do not directly "feed" the model knowledge itself, but teach it how to find, understand, and apply knowledge. Recently, several works Li et al. (2025b); Zheng et al. (2025b); Jin et al. (2025) have begun exploring training LLMs with a search engine to achieve adaptive interaction with the external knowledge for open-domain tasks.

However, these efforts primarily focus on evaluating model performance in so-called *closed-ended* or deterministic problems, which are typically defined by clear objectives and standard answers. Unlike closed-ended questions that seek precise answers, many real-world tasks involve *open-ended* or exploratory questions. As shown in Fig. 1, such questions typically lack a single, definitive answer, often requiring extensive, multi-turn search, yielding comprehensive responses that encompass multiple key findings. Given the intrinsic nature of these open-ended problems, developing effective methods to efficiently train and rigorously evaluate LLM-generated responses remains an active area of exploration.

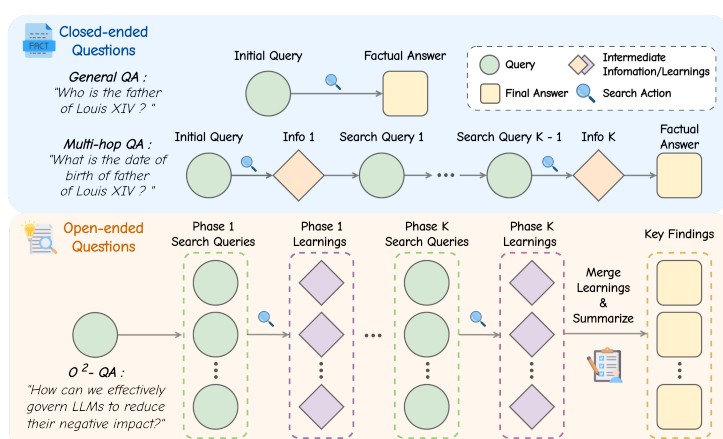

Figure 1: Illustration of different characteristics of closed-ended and open-ended questions.

While the open-ended and closed-ended questions have different formats and answer characteristics, solving them requires LLMs to combine external information retrieval with internal reasoning processes. To this end, we propose **O$^2$-Searcher**, a reinforcement learning (RL)-based **search** agent for the **open domain**, primarily focused on tackling demanding **open-ended** problems, yet also demonstrating strong performance on closed-ended ones. Specifically, to enable the O$^2$-Searcher to acquire external information rapidly and cost-effectively, we develop a search environment where the agent can freely explore. This local environment simulates an online search engine, returning several relevant cached web pages based on queries generated by the agent. To guide the agent's evolutionary trajectory, we design a series of reward functions to ensure the model simultaneously develops the capability to solve both open-ended and closed-ended problems.

Furthermore, we construct a manually curated open-domain open-ended question-answering (QA) benchmark, named O$^2$-QA, to effectively evaluate LLMs' performance on open-ended problems. Experimental results demonstrate that our O$^2$-Searcher, despite using only a small 3B parameter model, significantly surpasses the performance of state-of-the-art (SOTA) LLM agents on the O$^2$-QA benchmark. Moreover, on multiple closed-ended QA benchmarks, our O$^2$-Searcher not only achieves the SOTA performance with a comparable parameter size but also exhibits comparable performance against models with larger parameters. Our main contributions are summarized as follows:

- We introduce a novel RL-based search agent O$^2$-Searcher, which dynamically acquires and flexibly utilizes external knowledge via an efficient, local search environment. This design enables an effective decoupling of LLM's internal knowledge from its sophisticated reasoning processes.

- We propose a unified training mechanism, allowing the agent to efficiently handle both open-ended and closed-ended question types. Through meticulously designed reward functions, O$^2$-Searcher learns to identify problem types and adaptively adjust its answer generation strategies.

- We construct O$^2$-QA, a high-quality open-domain QA benchmark specifically designed to evaluate LLMs' performance on complex open-ended questions. It comprises 300 manually curated open-ended questions from diverse domains, along with ~30$k$ associated cached web pages.

- Extensive experiments show $O^2$-Searcher, which uses a 3B-base LLM, significantly outperforms other SOTA LLM agents on $O^2$-QA. It also achieves SOTA on multiple closed-ended QA benchmarks against comparably-sized models, with performance matching 7B models.

## 2 Related Works

### 2.1 LLMs with External Knowledge

LLMs, due to their reliance on parametric knowledge, inherently struggle with dynamic or evolving information, which often leads to factual inaccuracies Zhang et al. (2023); Peng et al. (2023); Ye et al. (2023); Zheng et al. (2025a); Kim et al. (2025). To mitigate these limitations and provide LLMs with timely, specific knowledge, integrating external knowledge has become a widely adopted strategy. This integration primarily occurs through two approaches: (1) Retrieval-Augmented Generation (RAG) Lewis et al. (2020); Asai et al. (2023); Yue et al. (2024), which enhances model output by retrieving relevant content via predefined workflows, though its fixed interaction patterns can limit adaptability; and (2) Search agent Trivedi et al. (2022a); Yao et al. (2023b); Alzubi et al. (2025); Jin et al. (2025); Song et al. (2025); Zheng et al. (2025b), where the LLM acts as an autonomous agent, deciding when and how to use a search engine and incorporate external knowledge. While RAG-based methods have achieved great advancement, their predefined procedures require extensive manual prompt engineering and limit generalization. Consequently, the search agent has gained traction, empowering LLMs to iteratively craft queries, invoke search engines, and generate results. Early methods Trivedi et al. (2022a); Yao et al. (2023b) relied on explicit instructions or engineered prompts. More recent progress, such as Search-R1 Jin et al. (2025) optimizing multi-turn queries and Deep-Researcher Zheng et al. (2025b) training LLMs for complex web navigation, focuses on enabling LLMs to adaptively learn search strategies via RL. Despite these advances, existing search agents remain focusing on closed-ended tasks and ignore open-ended exploration.

### 2.2 Advancing Reasoning in LLMs

Tackling open-domain problems requires models that deeply comprehend requirements and dynamically adjust strategies, making their reasoning capabilities crucial Patil (2025). There are three predominant approaches to enhance LLMs' reasoning ability: prompt-engineered, SFT-based, and RL-based methods. Prompt-engineered methods Wei et al. (2022); Wang et al. (2022b); Zhou et al. (2022); Yao et al. (2023a); Jiang et al. (2023); Zheng et al. (2025b) design prompts to provide exemplars or guidance that steer models along specific cognitive pathways, thereby improving problem-solving skills without modifying the underlying parameters. However, these methods do not alter the intrinsic model parameters and tend to produce rigid behavioral patterns, which can hinder instruction following, consistent reasoning, and generalization to novel tasks. Additionally, they require extensive manual prompt engineering to achieve reliable performance. SFT-based methods Xin et al. (2024); Ye et al. (2024); Fu et al. (2025) involve training models on large-scale, domain-specific reasoning datasets to instill reasoning abilities. While effective, this approach is costly in terms of data acquisition and can cause models to become overly sensitive to training data, limiting their generalization capabilities Cobbe et al. (2021). RL-based methods Sutton et al. (1999); Kaelbling et al. (1996); Shao et al. (2024); Yu et al. (2025); Cui et al. (2025); Liu et al. (2025b) guide models towards correct inferential steps via reward signals, offering high training efficiency and generalization potential; however, designing effective reward functions, especially for enabling continuous improvement in open-ended problems, remains a core challenge.

### 2.3 Agent Evolution

The evolution of LLMs from basic text generators to sophisticated autonomous agents marks a significant advancement in artificial intelligence Luo et al. (2025). Existing systems integrate multiple cognitive capabilities, including hierarchical planning Erdogan et al. (2025); Zhang et al.; Wang et al. (2023); Hu et al. (2023), tool manipulation Qiao et al. (2023); Yang et al. (2023); Yuan et al. (2024); Wu et al. (2024), and memory-augmented reasoning Yao et al. (2023b); Xi et al. (2025); Packer et al. (2023), enabling them to function as proactive problem-solvers. This progression has given rise to diverse architectural approaches, ranging from

single-agent implementations to complex Multi-Agent Systems (MAS) Hong et al. (2023); Li et al. (2023); Liang et al. (2025) where specialized LLM agents collaborate through emergent coordination protocols, albeit while introducing challenges in communication efficiency and conflict resolution. The demonstrated applications of these agentic systems span multiple domains, including software engineering Jimenez et al. (2023), web navigation tasks Zhou et al. (2023), and report generation OpenAI (2025); Google (2024). The evaluation of such capabilities has necessitated specialized benchmarks Qin et al. (2023); Valmeekam et al. (2023); Mialon et al. (2023) that assess not only task completion but also reasoning processes, operational efficiency, and safety considerations. In this work, we focus on the development of search agents, where LLMs are exploited to iteratively craft queries, invoke search engines, and generate answers. Different from existing methods Jin et al. (2025); Song et al. (2025); Zheng et al. (2025b) that only focus on closed-ended question answering tasks, our proposed method is capable of tackling both demanding open-ended and closed-ended problems.

## 3 Methodology

### 3.1 Overview

In this section, we present $O^2$-Searcher, capable of handling both open-ended and closed-ended questions in the open domain. $O^2$-Searcher interacts with the search environment to find, understand, and apply corresponding knowledge, potentially engaging in knowledge-aware multi-round interactions to generate the final answer. Let $q_0$ be the initial input query, $K_0$ represent an initial internal knowledge state, and $E$ be the search environment. In each round $t$, $O^2$-Searcher autonomously evaluates its current knowledge state $K_t$ and determines whether to formulate the final answer $a_{pred}$ or to continue gathering information for more knowledge. If more information is needed, it identifies knowledge gaps within $K_t$ and generate subsequent search queries $Q_t = G(K_t)$, interacts with $E$ to retrieve information $I_t = S(E, Q_t)$, and updates its internal knowledge state to $K_{t+1}$. This cycle of knowledge-aware assessment, targeted search, and knowledge update repeats until $K_t$ is deemed sufficient, at which point the final answer $a_{pred} = G(K_t)$ is generated and outputted.

### 3.2 Search Environment

While the open web provides a suitable dynamic search environment for open-domain question answering, practical applications in training scenarios are limited by slow API responses and high costs at scale. To address this, we develop a specialized search environment $E$ for our $O^2$-Searcher that efficiently supports discovery, processing, and utilization of external knowledge across various scenarios. This environment combines aggregated web pages for open-ended inquiries and structured Wikipedia-derived knowledge for closed-ended question answering. These heterogeneous sources are unified in a locally hosted knowledge corpus, enabling rapid information retrieval through dedicated tools followed by knowledge condensation via our condensation module.

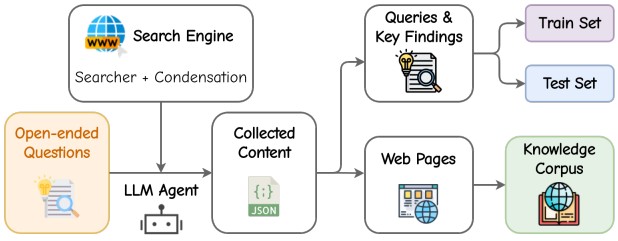

Figure 2: The construction of the knowledge corpus for open-ended questions.

**Knowledge corpus.** To train $O^2$-Searcher, we create a knowledge corpus for both open-ended exploration and closed-ended question answering. For open-ended tasks, we develop a specialized dataset comprising 300 expert-curated questions across cutting-edge domains (including AI, philosophy, sociology, energy transition, geopolitics, education, healthcare, bioethics, Web3, and metaverse). As shown in Fig. 2, each question undergoes systematic processing through a prompt-engineered LLM agent, leveraging commercial LLMs and the Serper API[1] for web retrieval. To model diverse retrieval behaviors, we execute randomized search trajectories per question. All retrieved content is cached to construct the retriever index. For closed-ended queries, we follow Jin et al. (2025) and locally index the 2018 Wikipedia dump Karpukhin et al. (2020) to

---

[1]https://serper.dev

---

**System**: SYSTEM_PROMPT
**User**: Initial query: How can intelligent manufacturing achieve sustainable industrial upgrading through innovative paths?
**Assistant**: <think>…it is necessary to first understand the current status of intelligent manufacturing…</think>
<search><query>The current status of intelligent manufacturing in various industries</query>
    <query>Definition and connotation of sustainable industrial upgrading</query> …
</search>
**User (Env_feedback)**: <learnings>…sustainable industrial upgrading involves the integration of advanced materials…</learnings>
**Assistant**: <think>Based on the information obtained,…</think>
<search><query>Development of new energy vehicles in recent years</query>
    <query>…</query> …
</search>
**User (Env_feedback)**: <learnings>…value of new energy vehicles…</learnings>
…
**Assistant**: <think>Based on the information obtained from multiple rounds of search…</think>
<answer>
- Sustainable industrial upgrading in intelligent manufacturing is supported by various innovative approaches, including the use of artificial intelligence, robotics, and automation.
- The adoption of innovative technologies in intelligent manufacturing plays a crucial role in promoting sustainable industrial upgrading, especially in reducing resource consumption.
…
</answer>

(a) open-ended question

---

**System**: SYSTEM_PROMPT
**User**: Initial query: when did Ford stop producing the 7.3 diesel?
**Assistant**: <think>To determine when Ford stopped producing the 7.3 diesel, we need to first understand the history of the 7.3 diesel engine in Ford vehicles.</think>
<search>< query>when did Ford stop producing the 7.3 diesel </query>
</search>
**User (Env_feedback)**: <learnings>…
    As the **7.3L V8** was no longer able to comply with emissions regulations for diesel engines, it was **discontinued midway through the 2003** model year. …
</learnings>
…
**Assistant**: <think>Now we know that Ford stopped producing the 7.3 diesel in 2003. Next, we can provide the answer.</think>
<answer>
2003
</answer>

(b) closed-ended question

Figure 3: We use multi-round conversations for modeling action trajectories to enhance interactivity. The agent reasons in <think> tags, searches via <search> (with specific queries in <query>), and answers in <answer> when ready. The search contents feedback by the search environment is presented in <learnings>. For the open-ended problem, multiple potential queries are generated.

facilitate efficient structured knowledge retrieval. By integrating these two types of content, we establish a specialized search environment $E$ with a locally hosted knowledge corpus, enabling $O^2$-Searcher to efficiently acquire external knowledge during execution process to generate answers in both closed-ended and open-ended question-solving.

**Retrieval tools.** In the context of open-ended retrieval, we utilize MeiliSearch[2] to query our curated web corpus, leveraging its high-throughput, low-latency performance, and flexible ranking mechanisms to closely emulate real-world web search dynamics. For closed-ended retrieval, we implement an E5-based dense retriever Wang et al. (2022a) over the 2018 Wikipedia dump Karpukhin et al. (2020), maintaining consistency with Jin et al. (2025) while benefiting from the high-precision passage ranking enabled by contrastively trained CCPairs embeddings. Both retrievers are configured to return the top-$k$ relevant passages per query based on their respective relevance scores. The passages retrieved for the given queries in the $t$-round interaction are combined as the retrieval information $I_t$.

**Information condensation.** The retrieval information $I_t$ for open-ended queries typically contains extensive raw web content, creating a significant computational burden in subsequent steps. Rather than directly incorporating raw contents into the knowledge state $K_{t+1}$, we introduce a condensation module that extracts structured learnings from $I_t$. Through prompting LLMs, the condensation module adaptively compresses input passages and extracts query-relevant items while maintaining semantic integrity. It employs length-aware compression: preserving most content with minimal restructuring for concise documents, while selectively retaining only salient information from extensive documents. This process eliminates redundant context while preserving query-relevant knowledge, reducing computational requirements, and improving reasoning efficiency.

## 3.3 Training Receipt

In this section, we elaborate on the training process of our $O^2$-Searcher for internalizing interaction and reasoning skills, enabling it to effectively find, understand, and apply relevant knowledge.

**Chat template.** As depicted in Fig. 3, we utilize a multi-round conversational methodology for modeling action trajectories to improve interactivity. The agent is instructed to conduct its reasoning within <think>

---

[2]https://www.meilisearch.com

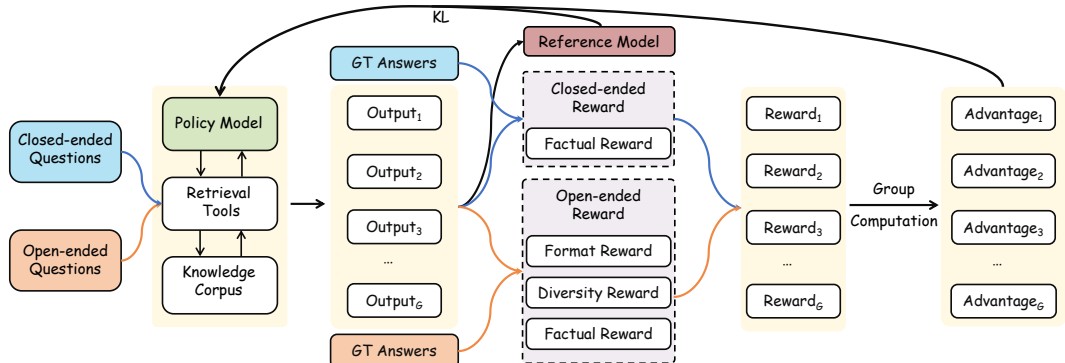

Figure 4: GRPO training with the interaction with the search environment. The policy model is optimized using GRPO with interaction with the local search environment, leveraging a reference model and rollout outputs from the preceding policy model. For closed-ended questions, optimization is driven by a Factual reward. For open-ended questions targeting key findings, training is guided by composite reward signals consisting of Format, Diversity, and Factual rewards.

tags prior to undertaking actions such as searching or answering. For information retrieval, the agent issues search actions encapsulated in <search> tags, which may include up to five distinct queries, each delimited by <query> tags. Subsequently, the retrieved data is presented within <learnings> tags. Ultimately, once the model deems the gathered information sufficient, it delivers its answer within <answer> tags.

**Cold start.** Drawing inspiration from DeepSeek-R1 Guo et al. (2025), we first apply the cold start to alleviate early instability and reduce the burden of learning the complex output format during RL training, especially for open-ended questions. This involves fine-tuning the instruction model on a small, curated dataset of CoT interaction trajectories to establish the initial RL actor. The data collection strategy varies by problem type. For open-ended questions, we employ the prompt-engineered LLM agent described in Section 3.2 to generate trajectories that capture detailed reasoning, search actions, retrieved data, and the final answer (e.g., key findings). For closed-ended questions, we apply Search-R1 Jin et al. (2025) to create example trajectories using small, randomly sampled subsets (∼1k samples) from the NQ Kwiatkowski et al. (2019) and HotpotQA Yang et al. (2018) training datasets.

**GRPO.** As shown in Fig. 4, we utilize the Group Relative Policy Optimization (GRPO) algorithm Shao et al. (2024) to reduce the computational resources required for RL training. For each input query in GRPO, a group of $G$ rollout trajectories, denoted as $\tau = \{y_i\}_{i=1}^G$, is generated using the preceding policy $\pi_{old}$, the current policy model $\pi_\theta$ is subsequently optimized by maximizing the objective function:

$$\mathcal{L}(\theta) = \mathbb{E}_{\substack{x \sim \mathcal{D}, \{y_i\}_{i=1}^G \\ y_i \sim \pi_{old}(\cdot|x)}} \left[ \frac{1}{G} \sum_{i=1}^G \min\left(r_i(\theta)A_i, \text{clip}(r_i(\theta), 1-\epsilon, 1+\epsilon)A_i\right) - \beta \mathbb{D}_{KL}[\pi_\theta || \pi_{ref}] \right] \tag{1}$$

where $\pi_{ref}$ denotes reference model, $r_i(\theta) = \frac{\pi_\theta(y_i|x)}{\pi_{old}(y_i|x)}$.

### 3.4 Reward Design

During the reinforcement training stage, we combine closed-ended questions with open-ended questions. Given the disparities in the format and characteristics of the answers for these two types of questions, we employ distinct reward functions to steer the training process.

Specifically, when dealing with closed-ended questions with exact answers, the training reward is directly correlated with the format correctness and accuracy of the generated answer. Following Jin et al. (2025), we only evaluate the final outcome and adopt the rule-based criteria as the factual reward, which is formulated as follows:

$$r_c = \begin{cases} \mathbb{I}(a_{pred} = a_{gt}), & \text{if format is correct} \\ 0, & \text{if format is incorrect} \end{cases} \tag{2}$$

where $a_{pred}$ and $a_{gt}$ are the predicted answer and ground truth, which are both converted to lowercase. $\mathbb{I}(\cdot)$ is the indicator function. Regarding the format, an answer is considered to be in the correct format when it is enclosed within <answer> tags.

For open-ended questions, we organize responses into a set of key findings, typically presented as long-form content that spans multiple aspects. Unlike closed-ended settings that require the model to match a single standard answer, our goal is to assess and enhance the model's ability to retrieve, integrate, and coherently organize relevant information in an open-ended context. Accordingly, we replace the notion of a single ground-truth answer with a reference information pool, which comprises a set of key findings extracted from the knowledge corpus within the local search environment. This formulation provides a more flexible basis for evaluating the quality of open-ended responses. To better guide the optimization process and generate high-quality answers in the desired format, we design a more sophisticated reward function. This function is composed of the following three parts:

**Format reward.** In addition to the requirement of adhering to the basic format where answers are enclosed within <answer> tags, we also expect the model to generate answers featuring diverse findings presented in Markdown list format. Consequently, answers with incorrect item formatting will be penalized. Moreover, to promote diversity among the items, we impose penalties for similarities and duplication between them. The detailed procedure is presented as follows:

$$r_{o,fm} = \alpha_0 \cdot (1 - \frac{n_{err}}{n_{tot}}) + \alpha_1 \cdot [1 - s(a_{pred})]^\delta - \alpha_2 \cdot (1 - n_{ind}/n_{tot}) \tag{3}$$

where $n_{tot}$ represents the total number of items generated, $n_{err}$ counts items with incorrect formatting, $n_{val}$ counts correctly formatted items ($n_{err} + n_{val} = n_{tot}$), $n_{ind}$ counts the unique items. Furthermore, $\alpha_0$, $\alpha_1$, $\alpha_2$ are non-negative weighting hyper-parameters (potentially constrained, e.g. $\alpha_1 + \alpha_2 = 1$, treating $\alpha_2$ as a separate penalty weight), and $\delta$ is an exponent. $s(a_{pred})$ evaluates the similarities among the finding items extracted from the generated answer $a_{pred}$:

$$s = w_0 \cdot \max(\mathbf{u}) + w_1 \cdot \operatorname{avg}(\mathbf{u}[\mathbf{u} > s_{thr}]) + w_2 \cdot \operatorname{avg}(\mathbf{u}) \tag{4}$$

where $\mathbf{u}$ represents the vector containing the pairwise similarity scores derived from the vector representations of the items, $s_{thr}$ is a threshold of potential highly similar pairs. The hyper-parameters $w_0$, $w_1$, and $w_2$ are non-negative, normalized weights.

**Diversity reward.** To encourage comprehensive information gathering through varied queries, we introduce a diversity reward $r_{o,div}$. This reward evaluates the semantic distinctiveness among the generated queries $Q = \{q_1, q_2, \ldots, q_{n_q}\}$ during the rollout process. First, we employ a pre-trained language model Wang et al. (2020) to encode each query $q_i$ into an embedding vector $\psi(q_i)$. We then compute the pairwise cosine similarities between all query embeddings, resulting in a query similarity matrix:

$$\mathbf{S}_{i,j}^q = \frac{\psi(q_i)^\top \psi(q_j)}{\|\psi(q_i)\| \cdot \|\psi(q_j)\|} \tag{5}$$

From this, we can derive a query independence matrix $\mathbf{T}$ where each element represents the dissimilarity between query pairs. The diversity score for each individual query $q_i$ is then calculated as its average independence score from all other queries:

$$s_i = \frac{1}{n_q - 1} \sum_{j \neq i} \mathbf{T}_{i,j}, \quad \mathbf{T}_{i,j} = 1 - \mathbf{S}_{i,j}^q \tag{6}$$

To encourage a balanced number of queries, i.e., avoiding both overly sparse and excessively redundant exploration, we introduce a weighting factor $\omega(n_q)$ that corresponds to the number of queries $n_q$. This factor penalizes deviations from an optimal range, thus alleviating the divergence Huang et al. (2025) without sacrificing breadth. The final diversity reward $r_{o,div}$ is computed as the average of these individual query scores, adjusted by the weighting factor:

$$r_{o,div} = \left( \frac{1}{n_q} \sum_{i=1}^{n_q} s_i \right) \cdot \omega(n_q) \tag{7}$$

Table 1: Results on open-ended $O^2$-QA benchmark. The best performance is set in **bold**. Our $O^2$-Searcher outperforms all baselines in the same local search environment, demonstrating the effectiveness of the RL training. In open web search, it maintains comparable performance to local search, highlighting the noise resilience.

| Method | Search Environment | F1 | | | LFS | | |
|---|---|---|---|---|---|---|---|
| | | Easy | Hard | Avg. | Easy | Hard | Avg. |
| Deepseek-v3 | Local | 0.0993 | 0.0798 | 0.0899 | 0.5119 | 0.2950 | 0.4070 |
| Doubao-32k | Local | 0.1167 | 0.0847 | 0.1012 | 0.5060 | 0.2916 | 0.4023 |
| Qwen2.5-72B | Local | 0.0978 | 0.0848 | 0.0915 | 0.4952 | 0.2735 | 0.3881 |
| GPT-4o-mini | Local | 0.1039 | 0.0779 | 0.0913 | 0.5296 | 0.3146 | 0.4257 |
| SFT | Local | 0.0651 | 0.0811 | 0.0728 | 0.1839 | 0.1940 | 0.1888 |
| SFT-Tool | Local | 0.0393 | 0.0385 | 0.0390 | 0.1898 | 0.1375 | 0.1654 |
| Search-R1 | Local | 0.0467 | 0.0317 | 0.0396 | 0.2443 | 0.1540 | 0.2009 |
| $O^2$-Searcher | Local | 0.2696 | **0.1743** | 0.2236 | 0.5956 | **0.3657** | 0.4845 |
| $O^2$-Searcher | Web | **0.2870** | 0.1691 | **0.2300** | **0.6301** | 0.3413 | **0.4905** |

where $n_q$ is the total number of queries generated.

**Factual reward.** To evaluate the factual correctness and information coverage of extracted finding items, we compute the F1 score by comparing the items derived from the predicted answer $a_{pred}$ against the reference answers $a_{ref}$ from the reference information pool. Specifically, we employ the same model Wang et al. (2020) as in the diversity reward to encode both sets of items into embedding vectors and compute their pairwise cosine similarities, yielding a similarity matrix that quantifies semantic alignment. Next, we derive a cost matrix from the similarity scores and apply the Hungarian algorithm Kuhn (1955) to establish optimal one-to-one correspondences between the generated and reference items. Pairs with similarity scores below a predefined threshold $s_\theta$ are discarded to ensure high-confidence matches. Finally, we compute precision, recall, and the aggregated F1 score based on the filtered pairs as the final factual reward $r_{o,f1}$. The procedure is formulated as follows:

$$\mathbf{S}^a_{i,j} = \frac{\phi(x_i)^\top \phi(y_j)}{\|\phi(x_i)\| \cdot \|\phi(y_j)\|} \tag{8}$$

$$\mathcal{M}' = \{(x_i, y_j) \mid \mathbf{M}_{i,j} = 1 \wedge \mathbf{S}^a_{i,j} \geq s_\theta\}, \quad \mathbf{M} = \text{Hungarian}(1 - \mathbf{S}^a) \tag{9}$$

$$p = |\mathcal{M}'|/n_{tot,pred}, \quad r = |\mathcal{M}'|/n_{tot,ref}, \quad r_{o,f1} = 2 \cdot p \cdot r/(p + r) \tag{10}$$

where $\phi$ is the embedding model, $x_i$ and $y_i$ are the items extracted from the generated answer and the reference answer. $n_{tot,pred}$ and $n_{tot,ref}$ stand for the total number of items in the generated set and the reference set, respectively.

Ultimately, for open-ended questions, if the generated answers are not enclosed by <answer> tags, the outcome reward is set to zero; or the outcome reward is calculated as the weighted sum of the format reward $r_{o,fm}$, diverse reward $r_{o,div}$ and the factual reward $r_{o,f1}$:

$$r_o = \gamma_0 \cdot r_{o,fm} + \gamma_1 \cdot r_{o,div} + \gamma_2 \cdot r_{o,f1} \tag{11}$$

where $\gamma_0$, $\gamma_1$, and $\gamma_2$ are non-negative hyper-parameters.

## 4 Experiments

We present the implementation details, benchmarks, main results, and analysis of $O^2$-Searcher in this section. Due to page limits, further experiments and details on the prompt and examples of application to report writing are provided in the Appendix.

Table 2: Exact Match (EM) metrics on closed-ended question-answering tasks. The best performance is set in **bold**. Our $O^2$-Searcher outperforms all baselines across most in/out-of-domain datasets using Qwen2.5-3B, achieving comparable performance to Search-R1-instruct (Qwen2.5-7B).

| Methods | In domain | | Out of domain | | | | | Avg. |
|---|---|---|---|---|---|---|---|---|
| | NQ | HotpotQA | TriviaQA | PopQA | 2wiki | Musique | Bamboogle | |
| **Qwen2.5-7B** | | | | | | | | |
| Direct Inference | 0.134 | 0.183 | 0.408 | 0.140 | 0.250 | 0.031 | 0.120 | 0.181 |
| CoT | 0.048 | 0.092 | 0.185 | 0.054 | 0.111 | 0.022 | 0.232 | 0.106 |
| IRCoT | 0.224 | 0.133 | 0.478 | 0.301 | 0.149 | 0.072 | 0.224 | 0.226 |
| Search-o1 | 0.151 | 0.187 | 0.443 | 0.131 | 0.176 | 0.058 | 0.296 | 0.206 |
| RAG | 0.349 | 0.299 | 0.585 | 0.392 | 0.235 | 0.058 | 0.208 | 0.304 |
| SFT | 0.318 | 0.217 | 0.354 | 0.121 | 0.259 | 0.066 | 0.112 | 0.207 |
| R1-base | 0.297 | 0.242 | 0.539 | 0.202 | 0.273 | 0.083 | 0.296 | 0.276 |
| R1-instruct | 0.270 | 0.237 | 0.537 | 0.199 | 0.292 | 0.072 | 0.293 | 0.271 |
| Search-R1-base | 0.480 | **0.433** | 0.638 | 0.457 | 0.382 | **0.196** | 0.432 | 0.431 |
| Search-R1-instruct | 0.393 | 0.370 | 0.610 | 0.397 | **0.414** | 0.146 | 0.368 | 0.385 |
| $O^2$-Searcher | **0.475** | 0.423 | **0.641** | **0.458** | 0.400 | 0.191 | **0.440** | **0.433** |
| **Qwen2.5-3B** | | | | | | | | |
| Direct Inference | 0.106 | 0.149 | 0.288 | 0.108 | 0.244 | 0.020 | 0.024 | 0.134 |
| CoT | 0.023 | 0.021 | 0.032 | 0.005 | 0.021 | 0.002 | 0.000 | 0.015 |
| IRCoT | 0.111 | 0.164 | 0.312 | 0.200 | 0.171 | 0.067 | 0.240 | 0.181 |
| Search-o1 | 0.238 | 0.221 | 0.472 | 0.262 | 0.218 | 0.054 | 0.320 | 0.255 |
| RAG | 0.348 | 0.255 | 0.544 | 0.387 | 0.226 | 0.047 | 0.080 | 0.270 |
| SFT | 0.249 | 0.186 | 0.292 | 0.104 | 0.248 | 0.044 | 0.112 | 0.176 |
| R1-base | 0.226 | 0.201 | 0.455 | 0.173 | 0.268 | 0.055 | 0.224 | 0.229 |
| R1-instruct | 0.210 | 0.208 | 0.449 | 0.171 | 0.275 | 0.060 | 0.192 | 0.224 |
| Search-R1-base | 0.406 | 0.284 | 0.587 | **0.435** | 0.273 | 0.049 | 0.088 | 0.303 |
| Search-R1-instruct | 0.341 | 0.324 | 0.545 | 0.378 | 0.319 | 0.103 | 0.264 | 0.325 |
| SFT-Tool | 0.371 | 0.258 | 0.503 | 0.387 | 0.186 | 0.103 | 0.192 | 0.286 |
| $O^2$-Searcher | **0.444** | **0.388** | **0.597** | 0.429 | **0.374** | **0.160** | **0.344** | **0.391** |

## 4.1 Implementation Details

We adopt Qwen2.5-3B-Instruct Yang et al. (2024) as the default backbone model in our proposed $O^2$-Searcher, and Qwen2.5-72B-Instruct as the default condensation module. For the stage of cold start, we utilize the Adam optimizer with an initial learning rate of $1 \times 10^{-5}$, warm-up ratio of 0.1, and a batch size of 16, training across 4 A100 GPUs for 2 epochs. During the RL training stage, we use the Verl framework [3]. At each training step, we sample 32 prompts per batch, each with 8 rollout trajectories, and optimize the policy model using the Adam optimizer with a reduced learning rate of $1 \times 10^{-6}$ and a mini-batch size of 32 on 4 A100 GPUs.

For the RL stage, the model was trained for 200 steps using a hybrid dataset of open-ended and closed-ended questions sourced from NQ Kwiatkowski et al. (2019) and HotpotQA Yang et al. (2018). For the GRPO training, the KL divergence regularization coefficient $\beta$ is set to 0.001, and the clip ratio $\epsilon$ is set to 0.2. We configure the maximum sequence length to be $10k$ tokens. Specifically, the retrieved content is also restricted to a maximum length of $2k$ tokens. The maximum searching step is set to 4. During the training procedure, we adopt vLLM [4] to accelerate LLM rollouts. The tensor parallel size is set to 1, and the GPU memory utilization ratio is set at 0.85. For rollout sampling, we use a temperature of 1.0 and a top-$p$ value of 1.0. The threshold $s_\theta$ is set to 0.75. Regarding the hyperparameters of the reward function, $\{\alpha_i\}_{i=0}^2 = \{0.5, 0.5, 3\}$, $\{w_i\}_{i=0}^2 = \{0.5, 0.3, 0.2\}$, $\{\gamma_i\}_{i=0}^2 = \{0.4, 0.4, 0.2\}$. $s_{thr}$ and $\delta$ are set to 0.6 and 1.5, respectively. These hyperparameters were mainly determined through empirical tuning. Specifically, in the

---

[3]https://github.com/volcengine/verl
[4]https://github.com/vllm-project/vllm

early stage of training, we tracked the value ranges and variation trends of different reward components and adjusted the hyperparameters to keep their magnitudes as comparable as possible. This helps avoid a single reward term dominating the optimization and ensures a better balance among different training objectives.

## 4.2 Benchmarks

**Datasets.** For the closed-ended question-answering task, we assess our proposed $O^2$-Searcher on both in-domain and out-of-domain datasets. The in-domain datasets include NQ Kwiatkowski et al. (2019) and HotpotQA Yang et al. (2018), while the out-of-domain datasets encompass TriviaQA Joshi et al. (2017), PopQA Mallen et al. (2022), 2WikiMultiHopQA Ho et al. (2020), Musique Trivedi et al. (2022b), and Bamboogle Press et al. (2022). In total, these validation tests involve 51,953 questions with corresponding ground-truth answers.

Regarding the open-ended evaluation, we construct an open-ended dataset, termed $O^2$-QA, derived from and intrinsically linked to the knowledge corpus developed in Sec. 3.2. $O^2$-QA consists of 300 expert-curated questions spanning a wide array of cutting-edge domains, including artificial intelligence, philosophy, sociology, energy transition, geopolitics, education, healthcare, bioethics, Web3, and metaverse. Among these, 240 questions are allocated for training, while 60 are reserved for testing. The test questions are further classified into easy and hard levels based on the median number of key findings in reference answers. The reference information pool for this dataset is derived directly from the content collected during the search process within the rollouts of the prompt-engineered LLM agents, as described in Sec. 3.2. Specifically, we apply DeepSeek-v3 Liu et al. (2024), Doubao-1.5-pro-32k [5], Qwen2.5-72B-Instruct Yang et al. (2024), and GPT-4o-mini Hurst et al. (2024) as the reasoning core of the LLM agents and employ GPT-4o to distill and synthesize key findings into concise reference answers.

**Baselines.** We follow the setting of Search-R1 Jin et al. (2025) for closed-ended question-answering tasks and compare our $O^2$-Searcher against three categories of methods: (1) CoT-based methods, including CoT Wei et al. (2022), RAG Lewis et al. (2020), IRCoT Trivedi et al. (2022a), and Search-o1 Li et al. (2025b). These methods leverage Chain-of-Thought reasoning either for direct inference or in combination with Retrieval-Augmented Generation (RAG). (2) SFT-based approaches, such as Supervised Fine-Tuning (SFT) Chung et al. (2024) and SFT-Tool. SFT does not involve interaction with a search engine, whereas SFT-Tool learns to output search actions by training on collected example trajectories, as described in the cold start stage in Sec. 3.3. (3) RL-based methods, including DeepSeek-R1 Guo et al. (2025) and Search-R1 Jin et al. (2025), both maintaining the same fundamental setting as Jin et al. (2025). DeepSeek-R1 performs reasoning and answer steps without search engines, whereas Search-R1 incorporates a local search engine.

For the open-ended task, we compare our $O^2$-Searcher with prompt-engineered LLM agents leveraging commercial LLMs such as Doubao-1.5-pro-32k, GPT-4o-mini, and open-source LLMs such as Deepseek-v3, Qwen2.5-72B-Instruct. We also include SFT-based baselines such as SFT and SFT-Tool, as well as the RL-based method Search-R1. In this scenario, SFT-Tool is trained on hybrid collected data containing both open-ended and closed-ended questions, while Search-R1 is prompted to produce key findings for open-ended evaluation.

**Metrics.** For closed-ended question-answering tasks, the Exact Match (EM) metric is applied, following Yu et al. (2024); Jin et al. (2025). For open-ended questions, evaluation relies on the F1 score (aligned with the RL training reward) and LLM-assessed Finding Similarity (LFS). Distinct from typical item-level, embedding-based F1 scores, LFS utilizes Qwen3.5-35B-A3B Team (2026) to assess semantic equivalence between entire generated and reference findings, yielding a dedicated semantic-level F1 score.

## 4.3 Main Results

**Open-ended benchmark.** Table 1 demonstrates that our $O^2$-Searcher achieves superior performance on both easy and hard levels of the open-ended $O^2$-QA benchmark when evaluated using F1 score and LFS metrics, outperforming all baseline methods operating within the constructed local search environment. Notably, our approach shows significant improvements over both SFT and SFT-tool variants, with perfor-

---

[5]https://seed.bytedance.com/en/special/doubao_1_5_pro

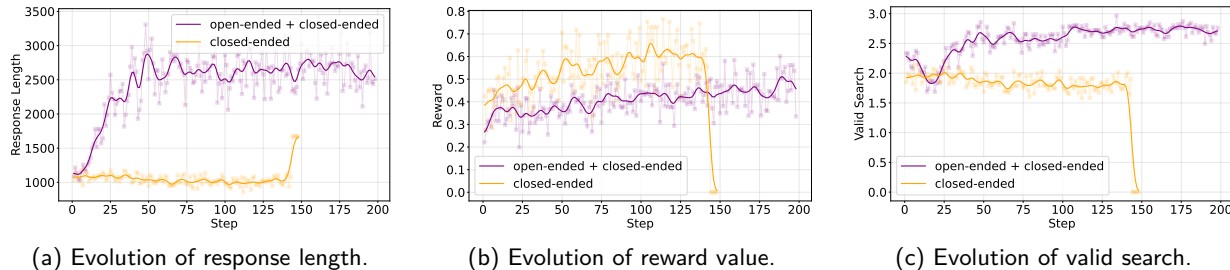

(a) Evolution of response length.     (b) Evolution of reward value.     (c) Evolution of valid search.

Figure 5: The evolution of response length, reward value, and valid search results across different training steps. Incorporating open-ended data yields superior training stability, longer average response lengths, and larger average search turns during the training procedure.

Table 3: Ablation on reward design. "IND" and "OOD" denote in-domain and out-of-domain.

| Reward components | Open-ended (F1) | | | Closed-ended (EM) | |
|---|---|---|---|---|---|
| | Easy | Hard | Avg. | IND Avg. | OOD Avg. |
| w/ $r_{o,f1}$ | 0.2169 | 0.1227 | 0.1714 | 0.407 | 0.371 |
| w/ $r_{o,fm}, r_{o,f1}$ | 0.2852 | 0.1678 | 0.2285 | 0.423 | 0.380 |
| w/ $r_{o,fm}, r_{o,div}, r_{o,f1}$ | **0.3104** | **0.1794** | **0.2523** | **0.434** | **0.393** |

mance margins that clearly validate the effectiveness of our RL training framework and the proposed reward functions. Furthermore, we evaluate $O^2$-Searcher in an open web search environment, where it maintains comparable performance to our constructed local search environment counterpart. This result highlights two key strengths of our method: (1) its ability to answer open-ended questions, and (2) its robustness in mitigating information noise inherent in open web environments. We also observe that using web search slightly degrades results on hard questions while providing a slight boost for easy ones. We attribute this to hard questions, which depend on identifying multiple key findings, are more susceptible to interference from the vast and often uncurated information available on the open web.

**Closed-ended benchmark.** Table 2 illustrates the results on closed-ended question-answering tasks. The results show that our $O^2$-Searcher demonstrates superior performance over all baselines across most in-domain and out-of-domain datasets when using the same Qwen2.5-3B backbone model. Notably, despite using only 1,200 closed-ended questions for RL training, our approach attains an average score on par with Search-R1-instruct (built on $\sim$ 190k samples and the larger Qwen2.5-7B-Instruct model). Furthermore, $O^2$-Searcher significantly outperforms SFT-Tool, which learns search capability solely through supervised fine-tuning on data in the cold start stage. This substantial performance gap underscores the necessity of RL training for robust search capability.

### 4.4 Analysis

**Training dynamics.** In Fig. 5, we depict the evolution of response length, reward value, and valid search results across different training steps during our unified training process, which includes both open-ended and closed-ended questions. The response length undergoes a rapid expansion in the early training stage (before step 50), followed by a slight vibration between steps 50 and 100. After step 100, it plateaus, suggesting stabilization in the model's output generation behavior. Meanwhile, the reward value demonstrates consistent improvement throughout training, with an initial sharp ascent followed by sustained growth. This trend indicates that the model progressively refines its reasoning abilities, optimizing its performance toward higher rewards as training advances.

We also present the training process of the model trained exclusively on closed-ended data in Fig. 5. Notably, in contrast to training solely on closed-ended data, incorporating open-ended data yields superior training stability, longer average response lengths, and larger average search turns during the training

procedure. Moreover, while models trained exclusively on closed-ended data exhibit a plateau or even a slight decline in valid search turns, our unified training approach reveals a more nuanced search behavior.

Specifically, valid search turns exhibit an initial decrease, followed by a gradual recovery and upward adjustment, before ultimately stabilizing. This observation indicates that unified training necessitates an adjustment of generation strategies between the two categories of questions. Consequently, the model is enabled to develop more sophisticated, context-sensitive generation strategies, as further substantiated by the results in Fig. 6.

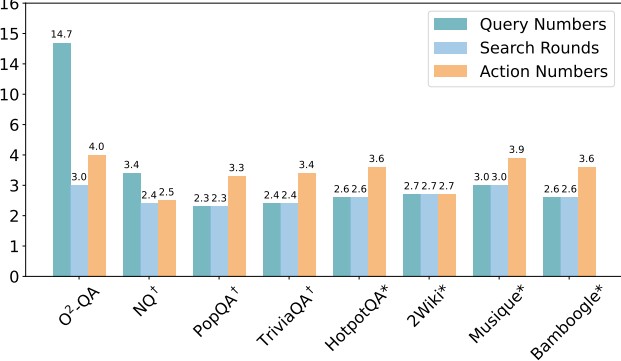

Figure 6: Search behavior across different datasets in the inference stage. [†] denotes general datasets, [*] denotes multi-hop datasets.

**Study of valid search on different datasets.** We further analyze the search behavior across different question types during the testing stage. As shown in Fig. 6, through the unified training of both open-ended and closed-ended data, our $O^2$-Searcher demonstrates several key characteristics: the number of search actions for multi-hop datasets is typically slightly higher than for general datasets, and the number of search actions for open-ended datasets is higher than for most closed-ended datasets. In terms of query complexity, the generated queries for open-ended problems are notably more extensive than those for closed-ended questions. This phenomenon indicates that our $O^2$-Searcher effectively learns to adapt its search behavior based on the specific characteristics of each question.

**Analysis of Reward Design.** Our analysis investigates the impact of different reward components in the open-ended task, with results presented in Table 3. All variants are trained for 300 steps in the RL training stage. These findings indicate that the choice of reward components significantly affects not only the open-ended task but also performance on closed-ended tasks. Notably, relying solely on the F1 score as the reward signal proves insufficient for guiding the learning process effectively, especially in open-ended scenarios. We also experimentally found that the use of only the F1 score can degrade the stability of the training, highlighting the importance of a more comprehensive design of reward for robust and effective model training.

**Case Study.** Fig. 8 depicts illustrative cases for the closed-end problem, generated by our $O^2$-Searcher, demonstrating the efficient search and answering capabilities of $O^2$-Searcher. We also provide a case for the open-ended problem, as shown in Fig. 9. We input an open-ended question, e.g., "how to learn a new language?", outside the training domain, and take the open web as the search environment. The results show that our $O^2$-Searcher can perform multi-run searching by generating multiple relevant queries to gather information and generate the final key findings for the open-ended question, demonstrating the generalization and adaptation ability to different types of questions.

## 5    Conclusion and Limitations

In this paper, we introduce $O^2$-Searcher, an RL-based search agent designed to decouple external information from its reasoning. $O^2$-Searcher dynamically acquires knowledge via a local simulated search environment and employs a unified training approach, enabling it to handle both open-ended and closed-ended questions by adapting its generation strategies. For robust evaluation on open-ended tasks, we constructed $O^2$-QA, a high-quality benchmark. Extensive experiments reveal that $O^2$-Searcher, even with a 3B parameter model, significantly surpasses existing LLM agents on $O^2$-QA and achieves SOTA performance on multiple closed-ended QA benchmarks against comparably-sized models. Our key limitations include $O^2$-QA's reliance on manually curated questions and a fixed web page cache, potentially introducing sampling bias, and the empirical validation being conducted solely on a 3B-parameter model, leaving scalability unevaluated. Dependence on external search also exposes models to noise and misinformation. As the first work exploring LLM evaluation and training on open-ended questions, this research lays the groundwork for developing more reliable, transparent, and grounded AI systems.

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

# A  Appendix

## A.1  Ablation on Condensation Module

In our original setup, we use Qwen2.5-72B-Instruct as the default condensation module. To better understand the impact of this design choice, we replace it with several alternative models and compare their performance. The results in Table 4 show that using different condensation models can affect downstream QA performance, and some stronger models achieve better results than the default Qwen2.5-72B-Instruct setting. In addition, we also examine whether increasing the amount of information produced by the condensation module is beneficial. Specifically, we double the amount of structured learnings in its output relative to the original setting. As shown in Table 4, generating more condensed learnings further improves performance, suggesting that the quality and quantity of the condensed intermediate output both play important roles in the final results.

Table 4: Comparison of different condensation modules. † denotes that more condensed learnings are output.

| Condensation Module | Easy | Hard | Avg. |
|---|---|---|---|
| GPT-4o-mini | 0.2868 | 0.1872 | 0.2386 |
| Gemini-2.5-flash | 0.2699 | 0.1900 | 0.2313 |
| Deepseek-v3 | 0.2826 | 0.1909 | 0.2383 |
| Qwen2.5-72B-Instruct | 0.2696 | 0.1743 | 0.2236 |
| Qwen2.5-72B-Instruct$^{\dagger}$ | 0.3216 | 0.1869 | 0.2565 |

## A.2  Generalization on Held-out Subset

In addition, to further evaluate the generalization ability of our method beyond the original 60 test questions, we construct an additional set of 75 manually verified, high-quality test questions. We adopt a reference-free "LLM-as-Judge" evaluation protocol to score and rank the outputs generated by different models. Specifically, for each test question, we collect the corresponding outputs (obtained by searching the open web) from all compared models and feed them into the judge model Qwen3.5-35B-A3B. The judge is asked to assign a score from 1 to 10 and produce a ranking based on the following criteria:

- **Relevance**: How well do the findings address the question?

- **Completeness**: Do the findings cover the key aspects of the topic?

- **Accuracy**: Are the findings factually correct and well grounded?

- **Depth & Insight**: Do the findings go beyond surface-level observations?

- **Clarity**: Are the findings clearly and concisely expressed?

We compare $O^2$-Searcher with DeepSeek-v3.2 Liu et al. (2025a), Gemini-2.5-flash Comanici et al. (2025), Doubao-1.5-pro-32k [6], Qwen2.5-72B-Instruct Yang et al. (2024), and GPT-4o-mini Hurst et al. (2024). This additional evaluation provides a complementary perspective on model generalization. As shown in Table 5, while $O^2$-Searcher demonstrates advantages over models such as Qwen2.5-72B and GPT-4o-mini, it still lags behind more recent frontier models such as DeepSeek-v3.2. We consider this a meaningful finding: for open-ended question answering, large-scale pretrained knowledge and model capacity remain key factors in determining answer quality.

---

[6]https://seed.bytedance.com/en/special/doubao_1_5_pro

Table 5: "LLM-as-judge" evaluation on a held-out set of manually verified open-ended questions.

|  | DeepSeek-v3.2 | Gemini-2.5-flash | $O^2$-Searcher | GPT-4o-mini | Qwen2.5-72B | Doubao-32k |
|---|---|---|---|---|---|---|
| Avg. Score | 8.88 | 8.20 | 7.99 | 7.73 | 7.64 | 7.43 |
| Avg. Rank | 1.97 | 3.20 | 3.57 | 3.81 | 4.13 | 4.31 |

### A.3 Discussion on Reward Hyperparameters

This section provides additional discussion on the hyperparameters used in the open-ended reward function. In our current implementation, the reward hyperparameters are selected through empirical tuning rather than exhaustive grid search, because each full RL run is computationally expensive. During the early stage of training, we monitor the value ranges and variation trends of different reward components and adjust the coefficients to keep their magnitudes comparable. This prevents a single reward term from dominating the optimization and encourages the policy to jointly learn answer formatting, non-redundant exploration, and factual coverage.

Table 6: Roles and sensitivity of the reward hyperparameters.

| Hyperparameters | Role | Sensitivity and practical effect |
|---|---|---|
| $\{\gamma_i\}_{i=0}^2$ | Balance the format, diversity, and factual rewards in the final open-ended reward. | These coefficients are the most important high-level knobs. In our experiments, the format reward is relatively easy to optimize and usually saturates earlier than the diversity and factual rewards. This indicates that, once the model has learned the required answer structure, moderate changes to $\gamma_0$ have limited marginal impact on final answer quality. We still keep a non-trivial $\gamma_0$ to provide a stable early-stage learning signal and prevent invalid outputs. In contrast, the diversity and factual rewards remain more informative after the format reward saturates, so their relative weights have a larger effect on search behavior. |
| $\{\alpha_i\}_{i=0}^2$, $\delta$ | Control the internal composition of the format reward, including valid item formatting, item-level semantic distinctiveness, and duplicated-item penalty. | These parameters are internal to the format reward, which typically saturates early in training. Once the model learns to produce valid structured answers, moderate changes to $\alpha_0$, $\alpha_1$, and $\delta$ have limited impact on final performance. We keep $\alpha_0$ and $\alpha_1$ comparable to encourage valid item formatting and item-level distinctiveness, while using a larger duplicate penalty $\alpha_2$ to discourage repeated findings. The exponent $\delta$ provides a smooth penalty for highly similar items. |
| $\{w_i\}_{i=0}^2$, $s_{thr}$ | Aggregate pairwise similarities among generated finding items when computing the redundancy score $s(a_{pred})$. | These parameters only affect the redundancy estimate inside $r_{o,fm}$, and their influence is further limited once the format reward saturates. Thus, moderate changes to $\{w_i\}_{i=0}^2$ and $s_{thr}$ are less likely to alter training than the diversity and factual reward weights. We assign the largest weight to the maximum similarity to detect near-duplicate findings, while average-based terms reduce sensitivity to noisy similarity pairs. The threshold $s_{thr}$ marks potentially redundant pairs under the embedding model and is transferable when using the same encoder. |

The component ablation in Table 3 also provides evidence about the impact of the reward design. Using only the factual F1 reward yields substantially weaker open-ended performance, while adding the format reward improves both open-ended and closed-ended results. Incorporating the diversity reward further improves performance, suggesting that the three components are complementary rather than redundant. This supports our design choice to use a balanced composite reward instead of relying on a single metric.

We expect the proposed choice to generalize across similar open-domain, open-ended QA settings for two reasons. First, the coefficients are tied to normalized quantities, such as ratios of valid items, cosine similarities, and F1 scores, rather than dataset-specific raw counts. Second, the tuning principle is component balancing: after changing the backbone model, embedding model, or answer format, one can reuse the same initial values and only inspect whether the component magnitudes remain comparable during early rollouts. In practice, retuning should focus on the high-level weights $\gamma_i$ and the semantic matching threshold $s_\theta$; the within-component redundancy weights are expected to require less adjustment as long as the same embedding model is used.

### A.4 Application of Writing Reports

The task of report writing represents a quintessential open-domain, open-ended problem, where crafting a superior report demands comprehensive referencing and robust empirical support for its assertions. $O^2$-Searcher is equipped with search capabilities to effectively gather and sift through pertinent online information, distilling key findings. This functionality significantly aids in the production of high-caliber reports. The collaborative process between $O^2$-Searcher and a designated writing agent for report generation is depicted in Fig. 7. The writing agent first generates an outline based on the key findings provided by our $O^2$-Searcher. Subsequently, it produces each section according to the respective outlines as well as the collected content and merges all sections to create the final report, complete with reference URLs.

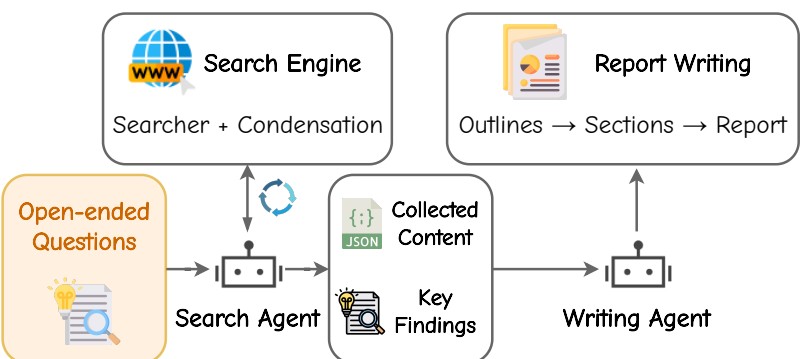

Figure 7: Illustration of the collaborative report writing process involving $O^2$-Searcher and the writing agent. $O^2$-Searcher retrieves, filters relevant information from the web based on a query, and summarizes key findings. Subsequently, the writing agent utilizes the output from $O^2$-Searcher to generate the report outline and the final document.

### A.5 Prompt Details

For the prompt-engineered LLM agent, we also apply the multi-round conversation format to generate the final key findings for open-ended problems. The system prompt is presented as follows:

**System prompt for LLM agents**

```
] ### Task Description
You are a preeminent expert researcher entrusted with a critical task.  Your goal is to identify
**key findings** related to the user's query by performing multi-round SEARCH actions and returning
an organized list.  Adhere to the following instructions:
```

```
### General Guidelines
- **User Context**:  Assume the user is highly knowledgeable and an experienced analyst; provide
responses with maximum detail, granularity, and accuracy without oversimplification.
- **Innovative Thinking**:  Explore unconventional ideas and insights to provide fresh perspectives,
beyond standard knowledge.
- **Proactive Anticipation**:  Predict what additional information might be relevant to the user and
incorporate it proactively in your research outcomes.
- **Accuracy Priority**:  Avoid errors at all costs; ensure that your results hold up under scrutiny
and are supported by evidence whenever possible.
- **Speculation**:  While speculation is allowed in cases where data may be incomplete, mark
speculative content clearly.
- **Focus on Key Findings**:  The final output must consist of well-synthesized, actionable, and
insightful findings directly addressing the user's query.
### Output Format
### SEARCH Action Execution:
When conducting SEARCH actions, enclose the search queries within <search> tags.  Use <query> tags to
specify each query.  Each query must be unique, with a maximum of 5 queries per round of SEARCH. The
format is as follows:
<think>THINKING PROCESS</think>
<search>
<query>QUERY 1</query>
<query>QUERY 2</query>
<query>QUERY 3</query>
<query>QUERY 4</query>
<query>QUERY 5</query>
</search>
### Final Key Findings Presentation:
Once you conclude your SEARCH actions, synthesize the information to produce actionable **key
findings** directly addressing the user's query.  Represent these findings as a flat JSON array,
encapsulated within <answer> tags.  Each array element should be a concise yet detailed finding,
ideally a single sentence or short paragraph.  The format is:
<think>THINKING PROCESS</think>
<answer>
[
"Key finding 1:  brief and precise insight.",
"Key finding 2:  brief and precise insight.",
......
"Key finding n:  brief and precise insight."
]
</answer>
### Key Requirements
1.  Conduct **multi-round SEARCH actions**, refining your queries over multiple iterations to gather
comprehensive, high-quality information.
2.  Limit the number of SEARCH rounds to a maximum of **5**.
3.  Craft highly **relevant and non-redundant findings** that focus on delivering maximum value and
insights to the user.
4.  Only proceed to the **final key findings presentation** once the SEARCH actions yield sufficient
information to address the user's request comprehensively.
### Examples
### SEARCH Action Execution Example:
<think>Begin with an initial search to collect broad background information and identify potential
sources for deeper exploration.</think>
<search>
<query>Impact of green energy policies on local economies</query>
<query>Recent advancements in solar panel technology and efficiency</query>
<query>Cost-effectiveness of implementing renewable energy solutions in urban areas</query>
</search>
### Final Key Findings Presentation Example:
<think>Synthesize insights from the SEARCH actions to formulate actionable key findings based on the
user's requirements.</think>
<answer>
[
"Key finding 1:  Green energy policies have led to a 15% increase in employment across renewable energy
sectors in 2023.",
"Key finding 2:  Solar panel efficiency improved by 22% due to innovations in perovskite-based designs
in recent years.",
```

```
"Key finding 3:  Urban renewable energy solutions yield a 25% reduction in overall energy costs for
mid-sized cities over 10 years.",
"Key finding 4:  Public-private partnerships are critical to overcoming initial capital barriers in
renewable energy projects."
]
</answer>
### Final Workflow
1.  Begin the task with an **analysis phase** to determine overarching themes and potential directions
for inquiry.
2.  Execute **multi-round SEARCH actions** to gather comprehensive data.
3.  Review, analyze, and synthesize the obtained information into **key findings**.
4.  Present the **key findings** in a flat JSON array structure as per the Final Key Findings
Presentation format.
```

When the LLM agent generates incorrect output (e.g., formatting errors), we use the following user prompt to guide correction:

**Error prompt for LLM agents**

```
The action you attempted before is invalid.  If you plan to execute actions like SEARCH, you need to
enclose the SEARCH queries within the <search> and </search> tags.  Furthermore, the required queries
for the SEARCH action should be placed between the <query> and </query> tags.  Moreover, if you wish to
present the final key findings for the initial query, you must wrap the result within the <answer> and
</answer> tags.
```

After retrieving new information from the search environment, we use the following prompt template to provide the LLM agent with condensed search results:

**Information prompt template for LLM agents**

```
After SEARCH action, we obtain learnings:  <learnings>LEARNING</learnings>.  Based on the context
within the conversation, you need to decide whether to execute an SEARCH action again for more
comprehensive information or output the final key findings according to the format specified in the
system prompt.  In the event that a SEARCH action is chosen, it is crucial to precisely delineate the
subsequent research directions according the context within the conversation.  Moreover, make sure that
you have gleaned sufficient insights through multiple rounds of executed SEARCH actions.  Only when
you are confident that you possess an ample amount of information to craft a thorough and detailed key
findings should you move forward with presenting the final key findings.
```

The system prompt for $O^2$-Searcher to handle both open-ended and closed-ended problems is presented as follows:

**System prompt for $O^2$-Searcher**

```
As an expert researcher, provide comprehensive key findings for open-ended queries and precise
answers to other specific questions.  Each time you receive new information, you MUST first engage
in reasoning within the <think> and </think> tags.  After reasoning, if you realize that you lack
certain knowledge, you can invoke a SEARCH action with distinct queries (one to five) using the
<search><query>QUERY</query><query>QUERY</query> </search> format to obtain relevant learnings, which
will be presented between the <learnings> and </learnings> tags.  You are allowed to perform searches
as many times as necessary.  If you determine that no additional external knowledge is required, you
can directly present the output within the <answer> and </answer> tags.
```

To compress the raw web content, we use the following prompt template:

**Prompt template for compressing raw web content**

```
You are an expert researcher.  Follow these instructions when responding:
- You may be asked to research subjects that is after your knowledge cutoff, assume the user is right
when presented with news.
- The user is a highly experienced analyst, no need to simplify it, be as detailed as possible and make
sure your response is correct.
- Be highly organized.
- Suggest solutions that I didn't think about.
```

```
- Be proactive and anticipate my needs.
- Treat me as an expert in all subject matter.
- Mistakes erode my trust, so be accurate and thorough.
- Provide detailed explanations, I'm comfortable with lots of detail.
- Value good arguments over authorities, the source is irrelevant.
- Consider new technologies and contrarian ideas, not just the conventional wisdom.
- You may use high levels of speculation or prediction, just flag it for me.
Given raw webpage contents:  <contents>CONTENTS</contents>, compress these contents into a **maximum
2K-token contents**, adhering to:
1.  Preserve critical information, logical flow, and essential data points
2.  Prioritize content relevance to the research query:
<query>QUERY</query>
3.  **Adjust length dynamically**:
- If original content < 2K tokens, maintain original token count ±10%
- If original content > 2K tokens, compress to ∼2k tokens
4.  Format output in clean Markdown without decorative elements
**Prohibited**:
- Adding content beyond source material
- Truncating mid-sentence to meet token limits
```

To extract the query-relevant learnings, we use the following prompt template:

**Prompt template for learning extraction**

```
Given the research query <query>QUERY</query>, your task is to extract a list of key learnings from the
provided contents.  Return a maximum of three distinct learnings.  If the contents are straightforward
and yield fewer insights, it's acceptable to return a shorter list.  Each learning should be unique,
avoiding any overlap or similarity with others.  Strive for conciseness while packing as much detailed
information as possible.  Be sure to incorporate any relevant entities such as people, places,
companies, products, or things, along with exact metrics, numbers, or dates.  These learnings will
serve as a foundation for further in-depth research on the topic.
<contents>CONTENTS</contents>
Generate the learnings as a single string, with each learning separated by a newline character.  Each
learning can consist of multiple sentences or a short paragraph.  Avoid numbering the learnings.
```

For the LFS calculation, we utilize a LLM to compare semantic equivalence between entire generated and reference findings and produce the matching results, the prompt template for this procedure is presented as follows:

**Prompt template for semantic matching**

```
You are a professional text similarity analysis expert.  Your task is to determine if input findings
are semantically similar to target findings.
Guidelines:
1.  You will receive two sets of findings:
- Input findings:  each separated by a newline
- Target findings:  each separated by a newline
2.  For each input finding, you need to:
- Analyze if it is semantically similar to any of the target findings
- If a similar entry is found, pair them together
- Each input finding can only be paired with one target finding
- Each target finding can only be paired with one input finding
3.  Output Requirements:
You need to output a list in JSON format, where each element is a pair:
[
["input finding 1", "matched target finding 1"],
["input finding 2", "matched target finding 2"],
...
]
Similarity Judgment Criteria:
1.  Core meanings should be identical or very close
2.  Even if expressions differ, pair them if core concepts match
3.  Partial overlap is not enough; main points must match
4.  If a finding contains multiple points, at least the main points must match
```

```
Please ensure your output follows the strict JSON format for subsequent processing.  Do not include any
explanatory text outside the JSON array.  If no matches are found, output an empty array [].

Input Findings List:   INPUT LIST
Target Findings List:   TARGET LIST
Please output the matching pairs strictly in the JSON format, without any additional explanatory text.
Important Notes:
1.  Output JSON array only, no additional explanatory text
2.  Use the complete original text for each finding
3.  If no matches are found, output an empty array []
```

The writing agent first generates an outline based on the key findings provided by our $O^2$-Searcher. Subsequently, it produces each section according to the respective outlines as well as the collected content and merges all sections to create the final report, complete with reference URLs. The prompt templates for the writing agent to generate outlines, sections, and a final report are presented as follows:

**Prompt template for outline generation**

```
You are an expert researcher.  Follow these instructions when responding:
- You may be asked to research subjects that is after your knowledge cutoff, assume the user is right
when presented with news.
- The user is a highly experienced analyst, no need to simplify it, be as detailed as possible and make
sure your response is correct.
- Be highly organized.
- Suggest solutions that I didn't think about.
- Be proactive and anticipate my needs.
- Treat me as an expert in all subject matter.
- Mistakes erode my trust, so be accurate and thorough.
- Provide detailed explanations, I'm comfortable with lots of detail.
- Value good arguments over authorities, the source is irrelevant.
- Consider new technologies and contrarian ideas, not just the conventional wisdom.
- You may use high levels of speculation or prediction, just flag it for me.

Given the following user query <query>QUERY</query> from the user.  Generate a report outlines to
systematically solve user's query.
Important!  Analysis requirements:
1.  First, carefully analyze the user's query to identify:
- The core question or problem the user is trying to solve
- Any specific aspects or dimensions they're particularly interested in
- The likely purpose of their research (practical application, theoretical understanding, etc.)
2.  Create a logical outline structure that:
- Starts with foundational concepts the user needs to understand
- Progresses through increasingly specific or complex aspects of the topic
- Concludes with practical applications or future implications relevant to the user's query
3.  Throughout the outline:
- Maintain consistent terminology aligned with the user's query
- Ensure each section directly contributes to answering the user's core question
- Avoid tangential information even if interesting but not directly relevant
The final outline should read as a single, unified document with a clear narrative arc that guides the
user from their initial question to a comprehensive understanding of the topic.
Here are distilled key findings relevent to user's query:  <contents>CONTENTS</contents>.
Organize the generated content entries as dictionary.  Don't use markdown code block label, just
represent the dictionary in JSON format.  Follow this operational protocol:
1.  Represent the outline as a dictionary where:
- Keys are chapters titles (e.g., "Introduction", "Analysis", "Conclusion").
- Values are lists of subsection titles or key points for each section.
2.  Create EXACTLY 5-7 first-level chapters (no more, no less).
3.  For each chapter, include 1-4 second-level sections that explore specific aspects (no more, no
less).
4.  Outline must Include only up to the SECOND level of the title, please DONOT write the third level
of the title or bullet points.
5.  ALL outline items should be writen in Chinese.
6.  Do NOT include any explanations, notes.  Don't use markdown code block label, just represent the
dictionary in JSON format.  Follow this operational protocol:
Example:
```

```
{ "Introduction":  { %ATTN: each chapter should be a dict not a list,
"Background":  "a concise description of this section in ONE sentence",
"Objective":  "a concise description of this section in ONE sentence",
....},
"Analysis":  {
"Key Findings":  "a concise description of this section in ONE sentence",
"Supporting Evidence":  "a concise description of this section in ONE sentence",
..},
"Conclusion":  {
"Summary":  "a concise description of this section in ONE sentence",
"Future Directions":  "a concise description of this section in ONE sentence",
...},
"FIRST_LEVEL_TITLE":  {
"SECOND_LEVEL_TITLE":  "a concise description of this section in ONE sentence",
"SECOND_LEVEL_TITLE":  "a concise description of this section in ONE sentence",
...},
}
```

**Prompt template for section generation**

```
Given the outlines <outlines>OUTLINES</outlines> of a specific section and the original contents
<contents>CONTENTS</contents>, write a detailed and coherent content based on the key points and
webpage content.  The outline is structured as a dictionary, where the keys represent section titles,
and the corresponding values are lists of subsection titles.  If a main section does not contain any
subsections, ensure the content is thorough and comprehensive, rather than limited to a few brief
sentences.
Include the following elements:
1.  Analysis:  Break down the information into meaningful insights.
2.  Synthesis:  Connect ideas from different sources to form a cohesive narrative.
3.  Explanations:  Provide clear and concise explanations of the content.
4.  Add in-text references (e.g., [1], [2]) based on the content index to reference the URLs where the
information was sourced.
5.  At the end of the section, list all the URLs referenced in the content.
Generate the contents using the Markdown format.
```

**Prompt template for final report generation**

```
Given the following initial query <query>QUERY</query> and contents <contents>CONTENTS</contents> of
each section from the user:
1.  Merge all sections into a single, cohesive report.  Try to retain the original content to maintain
the integrity and accuracy of the information.  Ensure the report is well-structured and flows
logically from one section to the next.
2.  Add in-text references (e.g., [1], [2]) to reference the URLs where the information was sourced.
3.  Include a References Section at the end of the report.  Provide a numbered list of the URLs
corresponding to the in-text citations.
4.  Generate the final report in Chinese, using Markdown format.  Follow these formatting guidelines:
- Use clear headings for main sections and subsections.
- Ensure the content is thorough and comprehensive, organized with paragraphs for each subsection
instead of listing items.
```

## A.6   Case Study of $O^2$-Searcher

Fig. 8 depicts illustrative cases for the closed-end problem, generated by our $O^2$-Searcher, demonstrating the efficient search and answering capabilities of $O^2$-Searcher. We also provide a case for the open-ended problem, as shown in Fig. 9. We input an open-ended question, e.g., "how to learn a new language?", outside the training domain, and take the open web as the search environment. The results show that our $O^2$-Searcher can perform multi-run searching by generating multiple relevant queries to gather information and generate the final key findings for the open-ended question, demonstrating the generalization and adaptation ability to different types of questions.

---

**System**: SYSTEM_PROMPT
**User**: Initial query: when did Ford stop producing the 7.3 diesel?
**O²-Searcher**: <think>To determine when Ford stopped producing the 7.3 diesel, we need to first understand the history of the 7.3 diesel engine in Ford vehicles.</think>
<search>< query>when did Ford stop producing the 7.3 diesel </query>
</search>
**User (Env_feedback)**:: Search learnings: <learnings>
- "Ford F-Series (seventh generation)" was offered alongside the Caterpillar 3208 V8; Caterpillar-powered vehicles were re-designated Ford "F-8000" (adding an extra "0" to the model name). During the late 1980s, the Caterpillar and Detroit Diesel engines are replaced by 6.6L and 7.8L inline-6 diesels produced in a joint venture between Ford and New Holland in Brazil. In 1992, the Ford-New Holland engines are both replaced by a Cummins 5.9L inline-6 diesel. Ford F-Series (seventh generation) The seventh generation of the Ford F-Series is a line of pickup trucks and medium-duty commercial trucks that was produced by Ford from 1979 to 1986. For the first time
- "Ford Super Duty" was sold with the 7.3L Power Stroke V8. Initially producing 235 hp/500 lb-ft of torque, the engine was retuned in 2001. Versions equipped with an automatic transmission produced 250 hp while manual-transmission examples produced 275 hp; with either transmission, the engine produced 525 lb-ft of torque. **As the 7.3L V8 was no longer able to comply with emissions regulations for diesel engines, it was discontinued midway through the 2003 model year.** As a running change during the 2003 model year, the 6.0L PowerStroke V8 was introduced as the replacement for the previous 7.3L V8 in LHD markets supplied with the
- "Ford Taurus" to discontinue the Taurus (along with the Fiesta, Focus, and Fusion), in order to focus more on its line of trucks and SUVs. The announcement was part of a plan by Ford Motor Company to cut costs and increase profits. This was in response to a shift in perceived consumer demand towards SUVs and pickup trucks, and away from sedans. On September 5, 2018, Ford ended all national and promotional advertising (including sales and special offers) for its entire sedan lineup, including the Taurus, whose production will end in 2019. A seventh generation of the Ford Taurus was introduced at
</learnings>
**O²-Searcher** : <think>Now we know that Ford stopped producing the 7.3 diesel in 2003. Next, we can provide the answer.</think>
<answer>
2003
</answer>

Figure 8: A closed-ended case produced by O²-Searcher

## A.7 Case of Report Writing

We present a case using the initial query: "Why can't Southeast Asia form a deep economic union like the EU?" Due to the extensive length of the generated report, we provide the key findings, outlines, and part of the generated report below:

---

**Key findings for the query "Why can't Southeast Asia form a deep economic union like the EU?"**

```
- Southeast Asia lacks the political unity and shared values necessary for forming a deep economic
union similar to the EU.
- The diverse cultures, languages, and economic structures of Southeast Asian countries make it
challenging to achieve a unified economic strategy.
- The geopolitical interests and rivalries among Southeast Asian countries hinder the formation of a
deep economic union.
- The complex political landscape and historical conflicts in Southeast Asia pose obstacles to
establishing a stable and cohesive economic union.
- The economic development levels and levels of industrialization differ significantly among Southeast
Asian countries, making it difficult to create a uniform economic policy.
- The existing political and economic frameworks in Southeast Asia, such as ASEAN, are not
well-developed and lack the institutional capacity to support a deep economic union.
- The economic policies and regulatory environments in Southeast Asian countries vary, which
complicates the process of forming a unified economic union.
- The lack of a common currency and monetary policy coordination in Southeast Asia further hinders the
formation of a deep economic union.
- The current state of globalization and regional integration in Southeast Asia is not conducive to the
creation of a deep economic union.
- The deep economic union model of the EU does not fully reflect the unique characteristics and needs
of Southeast Asian countries.
```

---

**Outlines for the query "Why can't Southeast Asia form a deep economic union like the EU?"**

```
# Introduction
- **Problem Presentation**:  Present the user's question about why Southeast Asia cannot form a deep
economic union similar to the EU
```

**System**: SYSTEM_PROMPT

**User**: Initial query: How to learn a new language ?

**O²-Searcher**: <think>To thoroughly explore the methods of learning a new language, it's necessary to understand the characteristics of language learning, the current status of the target language, personal learning habits, etc…</think>
<search>< query>Characteristics of language learning</query>
      < query>Current status of English language development </query>
      < query>Current status of Chinese language development</query>
      < query>Personal learning habits and cognitive preferences</query>
      < query>Commonly used language learning tools and resources</query>
</search>

**User (Env_feedback)**: Search learnings: <learnings>…various language projects achieved significant results, including the Rediscovering the Ancient Chinese Language Project, Digital Technology - Based Projects, Language Resources Protection Project, and Chinese Cultural Terminology Project.…</learnings>

**O²-Searcher** : <think>Based on the information obtained, we have learned about the current status of Chinese language education, the development of language learning tools and resources, and some basic concepts of language learning. Next, we need to analyze these pieces of information...</think>
<search>< query>Specific manifestations of visual, auditory, reading and writing, kinesthetic, verbal or linguistic, social or interpersonal, and solitary or intrapersonal learning styles</query>
      < query>Latest development of artificial intelligence technology and its application in language learning</query>
      < query>Future trends of big data technology in personalized learning</query>
      < query>Typical cases of corporate employee language training</query>
      < query>Development history and current situation of e-learning platforms</query>
</search>

**User (Env_feedback)**: Search learnings: <learnings>Visual learners process information best through seeing and observing, with a 2019 study claiming around 65% of people fall into this category…</learnings>

**O²-Searcher** : <think>Based on the information obtained from the first two rounds of searching, we have gained a deeper understanding of language learning, including learning styles, language learning tools and resources, and corporate language training...</think>
<search>< query>Comparison of different age groups' learning habits</query>
      < query>Application of blockchain technology in language learning</query>
      < query>Development history and current status of augmented reality technology</query>
      < query>Case studies of language learning in military and aviation industries</query>
      < query>Latest development of virtual reality technology</query>
</search>

**User (Env_feedback)**: Search learnings: <learnings>Children learn about the world through play before school and transition to a more focused learning process at school. They have difficulty concentrating, so lessons are typically 40 - 45 minutes long, and learning activities should engage multiple senses…</learnings>

**O²-Searcher** : <think>Based on the information obtained from multiple rounds of searching, we have gained a comprehensive understanding of language learning, including learning styles, language learning tools and resources, corporate language training, and emerging technologies. Next, we will present key findings about how to learn a new language, covering background, learning methods, challenges faced, and conclusions.</think>
<answer>
- Language learning is essential for skill development and career advancement, particularly in fields requiring cross-cultural communication and international collaboration.
- Learning a new language involves understanding the characteristics of language learning and adapting to different learning styles.
- Language learning tools and resources, such as apps and podcasts, can enhance learning effectiveness by providing interactive and immersive experiences.
- Personalized learning approaches, leveraging technology and data analytics, can improve language learning outcomes by addressing individual needs and preferences.
- The choice of language learning resources and methods should consider factors such as effectiveness, user experience, and cost.
- The current status of language learning is influenced by geopolitical factors, such as international educational initiatives and cultural exchange programs.
- Successful language learning requires a combination of theoretical knowledge, practical practice, and effective learning strategies.
…
</answer>

Figure 9: An open-ended case produced by our O²-Searcher.

```
- **Research Purpose**:  Explain that the purpose of this research is to understand the underlying
obstacles
# Overview of the EU's Deep Economic Union
- **Features of the Union**:  Introduce the characteristics of the EU's deep economic union in terms of
economic policies, currency, etc.
- **Formation Conditions**:  Analyze the political, economic and other conditions required for the EU
to form a deep economic union
# Internal Factors Hindering the Formation of a Deep Economic Union in Southeast Asia
- **Political and Value Differences**:  Point out the lack of political unity and shared values in
Southeast Asia
- **Diversity of Economic Structures**:  Illustrate the significant differences in economic development
levels and industrialization degrees among countries
- **Differences in Policies and Regulations**:  Emphasize the obstacles brought by different economic
policies and regulatory environments
# External Factors Hindering the Formation of a Deep Economic Union in Southeast Asia
- **Geopolitical Influence**:  Explain how geopolitical interests and rivalries impede the formation of
the union
- **Legacy of Historical Conflicts**:  Describe the impact of historical conflicts on establishing a
stable economic union
# Analysis of Existing Frameworks and the Globalization Background
- **Limitations of Existing Frameworks**:  Analyze the deficiencies of existing frameworks such as
ASEAN in supporting a deep economic union
- **Current Situation of Globalization and Regional Integration**:  Explore the unfavorable impact of
the current globalization and regional integration situation
# Conclusion and Outlook
- **Summary**:  Summarize the main reasons why Southeast Asia cannot form a deep economic union
- **Future Directions**:  Propose possible directions for promoting economic integration in Southeast
Asia in the future
```

**Part of report for the query "Why can't Southeast Asia form a deep economic union like the EU?"**

```
# Why can't Southeast Asia form a deep economic union like the EU?
## Introduction
### Problem Statement
The central question is why Southeast Asia struggles to form a deep economic union comparable to the
European Union (EU). This exploration aims to understand the underlying obstacles preventing Southeast
Asia from achieving such a high level of economic integration.
### Research Purpose
The purpose of this study is to comprehensively analyze and explain the fundamental barriers hindering
the formation of a deep economic union in Southeast Asia, similar to that of the EU. By doing so, we
can better understand the unique challenges and complexities of the Southeast Asian region.
...
## Conclusion and Outlook
### Summary:  Main Reasons Why Southeast Asia Cannot Form a Deep Economic Union
#### Economic Disparities
Southeast Asian countries exhibit significant economic disparities, evident from GDP and GDP per capita
data...
#### Political and Geopolitical Challenges
- **Territorial Disputes**:  The South China Sea is a major source of tension among Southeast Asian
countries and between them and China..
- **External Influences**:  The region is caught in the geopolitical rivalry between the United States
and China...
#### Cultural and Social Differences
Southeast Asia is a region with diverse cultures, religions, and languages [4].  Major religions
include Buddhism, Islam, Hinduism, Christianity, Confucianism, Taoism, and Animism, each with its
own set of beliefs and practices...
#### Regulatory and Institutional Barriers
- **Lack of Harmonization**:  Regulatory frameworks in Southeast Asian countries are not fully
harmonized...
- **Weak Institutional Capacity**:  Some ASEAN countries may have weak institutional capacity in terms
of policy implementation, governance, and regulatory enforcement...
### Future Directions for Promoting Economic Integration in Southeast Asia
#### Bridging Economic Gaps
- **Infrastructure Development**:  Investing in regional infrastructure projects can help bridge
economic gaps.  For example, developing transportation networks such as roads, railways, and ports
can improve connectivity between less developed and more developed areas.  This can facilitate trade,
```

```
attract investment, and promote economic growth in less developed regions...
- **Capacity Building Programs**:  Implementing capacity building programs in areas such as education,
skills training, and technology transfer can help less developed countries catch up with more advanced
ones...
#### Addressing Political and Geopolitical Issues
- **Strengthening Conflict Resolution Mechanisms**:  ASEAN should strengthen its conflict resolution
mechanisms, particularly with regard to territorial disputes...
- **Balancing External Relations**:  Southeast Asian countries should continue to balance their
external relations and avoid over-reliance on any single major power...
#### Leveraging Cultural and Social Diversity
- **Cultural Exchange and Understanding**:  Promoting cultural exchange and understanding among
Southeast Asian countries can help overcome cultural barriers to economic integration...
- **Catering to Diverse Consumer Needs**:  Businesses can leverage the cultural and social diversity of
the region by developing products and services that cater to different consumer preferences...
#### Strengthening Regulatory and Institutional Frameworks
- **Regulatory Harmonization**:  ASEAN should pursue greater regulatory harmonization in key areas such
as trade, investment, intellectual property, and the digital economy...
- **Institutional Strengthening**:  Strengthening ASEAN's institutional capacity is crucial for the
effective implementation of economic integration initiatives...
```

