# OpenReview forum: "O$^2$-Searcher: A Searching-based Agent Model for Open-Domain Open-Ended Question Answering"
_TMLR — Accepted by TMLR_

### Review · Reviewer_VghA · 2026-03-12

**Summary Of Contributions:**

This paper introduced O2-Searcher, an agentic paradigm based on RL to solve both open-ended and close-ended questions in the open domain.

Strengths
- Optimizes for an important domain (agent-based open- and close-ended QA)
- Introduces O2-QA, a benchmark for open-ended QA

Weaknesses
- Reliance on intricate hyperparameter choices in the reward function
- Evaluation setup: small benchmark size, reliance on proprietary models for evaluation metrics, potential artificial advantage in the search environment

**Audience:**

Yes

**Audience Explanation:**

- RL post-training: Applying RL techniques that are similar to GRPO for tool-use and searching is interesting and relevant
- Agentic applications: Developing LLMs that can invoke tools to access external knowledge is a timely topic

**Claims And Evidence:**

Yes

**Claims Explanation:**

Overall, the claims are supported by solid experimental evidence. However, there are a few areas where improvements would strengthen the paper:
- Evaluations for open-ended questions. Only 60 questions are in the test set, and the main results do not include statistical signals like confidence intervals, making it difficult to appreciate the groundness of the results.
- The local search environment. The environment for the open-ended task is built by running a prompt-engineered agent on 300 questions and caching the retrieved pages. This feels slightly arbitrary. What if the retrieved content does not contain the "ground truth" needed to answer a query? There is a possibility that O2-Searcher has an artificial advantage during evaluations because the search environment has been pre-populated with relevant documents for those exact queries.
- No sensitivity analysis of the hyperparameters in the reward function, which undermines the generality of this approach.
- I commend the authors for including ablations on the reward design

**Requested Changes:**

Critical to securing my recommendation for acceptance
- Address the small test set size. Only 60 questions are used for testing to produce the results. It's crucial that the authors provide statistical significance testing (e.g., confidence intervals) to demonstrate that O2-Searcher's performance gains are not due to noise.
- Clarify the evaluation setup. Why did the authors not evaluate baseline methods and models in the open web environment?
- Sensitivity analysis of hyperparameters. The reward function introduces several hyperparameters that seem to have been tuned meticulously ("{αi} = {0.5, 0.5, 3}, {wi} = {0.5, 0.3, 0.2}, {γi} = {0.4, 0.4, 0.2}. sthr and δ are set to 0.6 and 1.5"). Adding a brief sensitivity analysis on how these parameters affect the model quality would strengthen the evaluations and show generality of the approach. At the very least, a discussion should be added on how these values are derived.
- Justify the LFS metric. Why did the authors pick Doubao-1.5-pro-32k for the LFS metric? Without information on its parameter count, it's unclear whether this metric is trustworthy. I recommend that the authors rerun this evaluation using an open-source model so future work could attempt to replicate the metric.
- Justify the format reward in the reward function. It feels weird that the work assumes that all open-ended questions are best answered with a markdown list of findings. I asked ChatGPT to generate this example question: "How did your perspective on research change during your PhD?", which is by definition an open-ended question that doesn't necessarily rely on a list.

Minor comments and feedback
- Typo in 3.4: "we combine closed-ended questions with closed-ended questions"
- Why is this work called O2-Searcher? Is there any significance behind the name?

---

> ### Author Response · Authors · 2026-04-01
> **Response to the reviewer:part1**
>
> Dear Reviewer VghA:
>
> We sincerely thank you for your thoughtful feedback and for acknowledging the contributions of our work. We truly appreciate the opportunity to engage in a deeper discussion on the points you raised.
>
> **Q1**: It's crucial that the authors provide statistical significance testing (e.g., confidence intervals) to demonstrate that O2-Searcher's performance gains are not due to noise.
>
> **A1**: Thank you for raising this concern. To assess whether the performance gains of O2-Searcher could be attributed to randomness or noise, we conducted 5 independent rollouts on the original 60-question test set and report the mean and standard deviation of the results. As shown below, the standard deviations are relatively small across all settings, suggesting that the model outputs are stable. These observations indicate that the improvements of our method are unlikely to be caused by noise.
>
> |       | Mean | Std |
> |-------|------|-----|
> | Avg. | 0.23434 | 0.01269 |
> | Hard | 0.18047 | 0.01340 |
> | Easy | 0.28471 | 0.01427 |
>
> In addition, to further evaluate the generalization ability of our method beyond the original 60 questions, we constructed an additional set of 75 manually verified, high-quality test questions.
>
> Following the suggestion for review juij, we adopted a reference-free “LLM-as-Judge” evaluation protocol to score and rank the outputs generated by different models. Specifically, we fed each test question together with the corresponding outputs (obtained by searching the open web) from all compared models into the judge model Qwen3.5-35B-A3B, which was asked to assign a score from 1 to 10 and provide a ranking based on the following criteria:
>
> - **Relevance**: How well do the findings address the question?
> - **Completeness**: Do the findings cover the key aspects of the topic?
> - **Accuracy**: Are the findings factually correct and well grounded?
> - **Depth & Insight**: Do the findings go beyond surface-level observations?
> - **Clarity**: Are the findings clearly and concisely expressed?
>
> We evaluated DeepSeek-v3.2, Gemini-2.5-flash, GPT-4o-mini, Qwen2.5-72B-Instruct, Doubao-1.5-pro-32k, and our O2-Searcher on this held-out subset. The results are summarized in the table below.
>
> | Model              | Avg. Score | Avg. Ranking |
> |-------------------|----------:|------------:|
> | DeepSeek-v3.2     | 8.88      | 1.97        |
> | Gemini-2.5-Flash  | 8.20      | 3.20        |
> | O2-Searcher       | 7.99      | 3.57        |
> | GPT-4o-mini       | 7.73      | 3.81        |
> | Qwen2.5-72B       | 7.64      | 4.13        |
> | Doubao-32k        | 7.43      | 4.31        |
>
> This additional evaluation provides a complementary view of model generalization. Although O2-Searcher shows an advantage over models such as Qwen2.5-72B and GPT-4o-mini, it still lags behind more recent frontier models such as DeepSeek-v3.2. We explain this is because, for open-ended QA, extensive pretraining knowledge and larger model scale still play an important role in determining answer quality.
>
> We have clarified these points in the revision.
>
> **Q2**: Why did the authors not evaluate baseline methods and models in the open web environment?
>
> **A2**: Thank you for this question. We did not conduct a unified evaluation of all baseline methods and models in the open-web environment in the original submission for two main reasons. First, when we prepared the original submission, due to the practical constraints of search API availability, access conditions, and query resource limits, it was difficult to support open-web retrieval uniformly for all compared methods and models. Second, the local search environment we constructed provides a **consistent, stable, and reproducible information space** for all methods. This setup allows us to more accurately isolate and compare each model’s ability in information retrieval, integration, and organization, without introducing additional variability from external web search systems.
>
> **Q3**: Adding a brief sensitivity analysis on how these parameters affect the model quality would strengthen the evaluations and show the generality of the approach. At the very least, a discussion should be added on how these values are derived.
>
> **A3**: Thank you for pointing this out. We agree that the reward function introduces several hyperparameters, and clarifying how they were chosen is important. In our current implementation, these hyperparameters were selected primarily through **empirical tuning**. More specifically, during the early stages of training, we monitored the numerical ranges and variation trends of the different reward components and adjusted the hyperparameters so that the magnitudes of the reward terms were kept as close as possible. This was done to prevent any single reward component from dominating the optimization process and to better balance the contributions of the different training objectives.
>
> We have clarified these points in the revision.

---

> > ### Author Response · Authors · 2026-04-01
> > **Response to the reviewer: part2**
> >
> > **Q4**: I recommend that the authors rerun this evaluation using an open-source model so future work could attempt to replicate the metric.
> >
> > **A4**: Thank you for this valuable suggestion. We initially chose Doubao-1.5-pro-32k as the evaluator for the LFS metric because, in our preliminary experiments, it demonstrated relatively strong instruction-following ability and produced stable judgment results across repeated evaluations.
> >
> > At the same time, we fully agree with the reviewer’s concern regarding trustworthiness and reproducibility. Following the suggestion, we additionally conducted the LFS evaluation using the open-source model **Qwen3.5-35B-A3B**, with the final score obtained by averaging over three runs. The results show that the conclusions remain largely consistent with those obtained using Doubao-1.5-pro-32k, especially in terms of the overall ranking trend and the relative advantage of O2-Searcher.
> >
> > The results are shown below:
> >
> > | Model | Search Environment | Easy | Hard | Avg |
> > |-------|--------------------|------|------|-----|
> > | DeepSeek-v3 | local | 0.5119 | 0.2950 | 0.4070 |
> > | Doubao-32k | local | 0.5060 | 0.2916 | 0.4023 |
> > | Qwen2.5-72B | local | 0.4952 | 0.2735 | 0.3881 |
> > | GPT-4o-mini | local | 0.5296 | 0.3146 | 0.4257 |
> > | SFT | local | 0.1839 | 0.1940 | 0.1888 |
> > | SFT-Tool | local | 0.1898 | 0.1375 | 0.1654 |
> > | Search-R1 | local | 0.2443 | 0.1540 | 0.2009 |
> > | O2-Searcher | local | 0.5956 | 0.3657 | 0.4845 |
> > | O2-Searcher | web | 0.6301 | 0.3413 | 0.4905 |
> >
> > We have updated the results in the revision.
> >
> > **Q5**:  It feels weird that the work assumes that all open-ended questions are best answered with a markdown list of findings.
> >
> > **A5**: Thank you for raising this point. We fully agree that for many open-ended questions, a narrative response may be more natural and appropriate. We would like to clarify that the format reward in our paper is not intended to claim that a markdown list is the universally best response format for all open-ended questions. Rather, it is introduced for the specific purposes of task formulation and evaluation standardization.
> >
> > In our setting, we encourage the model to present its answer as a list of key findings, mainly for two reasons:
> >
> > 1. **Facilitating evaluation.**
> >    A structured list helps decompose the model output into relatively clear information units, which makes it easier to conduct a semantic-level coverage assessment.
> >
> > 2. **Better aligning with the target task.**
> >    The focus of this work is on open-ended information seeking, aggregation, and organization. For this type of task, a structured format is often helpful for presenting the core facts, evidence, and viewpoints gathered from retrieval and reasoning.
> >
> >
> > **Q6**: Why is this work called O2-Searcher?
> >
> > **A6**: Thank you for your interest. The name O2-Searcher refers to a search-oriented model for **open-ended** and **open-domain** question answering.
> >
> > **Q7**: typo erros.
> >
> > **A7**: Thank you for your careful reading. We have proofread the paper and corrected the corresponding typos.

---

### Review · Reviewer_juij · 2026-03-17

**Summary Of Contributions:**

This paper proposes $O^2$-Searcher, an RL-based search agent that uses GRPO with tailored reward functions to handle both open-ended and closed-ended questions in open-domain settings. It introduces a local search environment combining cached web pages and Wikipedia for efficient training, and designs a composite reward (format, diversity, factual) to guide open-ended answer generation. The authors also construct $O^2$-QA, a 300-question benchmark for evaluating open-ended QA. Experiments show the 3B-parameter agent outperforms larger models on $O^2$-QA and achieves competitive closed-ended QA results.

**Audience:**

Yes

**Audience Explanation:**

I believe that researchers in the TMLR community studying reinforcement learning, search agents, and open-ended question answering will find the ideas presented in this paper of interest.

**Broader Impact Concerns:**

N/A for this paper.

**Claims And Evidence:**

No

**Claims Explanation:**

The overall structure of the paper is sound, featuring extensive experimentation and theoretical proofs, however, I harbor doubts regarding certain sections and remain unconvinced by them.

1. The paper consistently fails to disclose the specific Large Language Models (LLMs) used to construct the corpus and generate the "ground-truth" answers, referring to them only vaguely as "commercial LLMs." This poses a significant challenge to the reproducibility of the work. Furthermore, the knowledge corpus used for training and the ground-truth answers for the test set were both derived from the search trajectories of the same agent, after which commercial LLMs were employed to distill these into "standard answers." This makes it difficult to convincingly demonstrate that the model genuinely possesses the capability to answer open-ended questions.

2. A core characteristic of open-ended questions is the "absence of a standard answer"; yet, the F1 and LFS metrics utilized in this paper rely entirely on matching against ground-truth references. This fundamentally contradicts the very nature of open-ended questions. Although the paper itself defines open-ended questions in Section 1 as "lacking a standard answer," its evaluation framework is entirely predicated on "standard answer matching."

3. In the open-ended evaluation presented in Table 1, baseline models such as DeepSeek-V3 and GPT-4o-mini were evaluated using a "prompt-engineered" approach; they underwent no specific fine-tuning to adapt to the O²-QA format (specifically, the list of "key findings"). This disparity in methodology may result in an unfair comparison.

4. The dataset comprises a mere 300 questions (240 for training and 60 for testing); with a test set consisting of only 60 samples, it is statistically difficult to draw reliable conclusions. Moreover, while the paper claims to cover a diverse range of fields including AI, philosophy, and sociology, distributing these 60 test samples across so many domains likely leaves each individual field represented by only a handful of questions.

5. The beginning of Section 3.4 states, "we combine closed-ended questions with closed-ended questions"; this should be "open-ended questions with closed-ended questions."

**Requested Changes:**

First, this paper employs a unified framework to simultaneously address two distinct types of problems is conceptually significant. The results for closed-ended tasks are particularly strong, with the 3B model achieving performance levels comparable to a 7B model. The design philosophy behind the reward function (incorporating format, diversity, and factual accuracy) offers valuable insights.

I recommend that the authors incorporate reference-free "LLM-as-Judge" evaluations or human assessments to validate performance on open-ended tasks; expand the scale of the O²-QA dataset and publicly release the details of its construction; ensure fair format adaptation for baseline models.

---

> ### Author Response · Authors · 2026-04-01
> **Response to the reviewer: part1**
>
> Dear Reviewer juij:
>
> We sincerely thank the reviewer for their thorough evaluation and constructive feedback. We are greatly encouraged that you highlighted our primary contributions. In the following, we provide detailed discussions and explanations regarding the concerns raised.
>
> **Q1**: The specific LLMs used to construct the corpus and generate the "ground-truth" answers. It is difficult to convincingly demonstrate that the model genuinely possesses the capability to answer open-ended questions.
>
> **A1**: We thank the reviewer for raising this important concern. First, we clarify the specific models used in data construction and reference information generation. The commercial LLMs for random rollouts include DeepSeek-v3, Doubao-1.5-pro-32K, Qwen2.5-72B-Instruct, and GPT-4o-mini. In addition, the model for generating the reference information pool is GPT-4o, and the condensation model is set to Qwen2.5-72B-Instruct by default.
>
> At the same time, we would like to further clarify the design intention of our benchmark. Our goal is **not** to evaluate whether a model can reproduce a single “standard answer.” Instead, we aim to assess whether a model can **retrieve, integrate, and organize relevant information** for open-ended questions. Accordingly, our evaluation target is **not a single ground-truth answer**, but rather a **reference information pool**.
>
> More specifically, for each topic, we use multiple different LLMs to independently search for and synthesize relevant information, and then aggregate their outputs into a reference information pool. This design serves two purposes:
>
> 1. **Improving coverage**: to include as much important topic-relevant information as possible.
> 2. **Reducing single-generator bias**: to avoid having the “standard answer” determined by the blind spots or stylistic preferences of any one model.
>
> Therefore, what we evaluate is not whether the model matches one distilled answer, but whether it can recover the key information contained in the reference information pool under an open-ended setting.
>
> We have clarified these points in the revision.
>
> **Q2**: A core characteristic of open-ended questions is the "absence of a standard answer"; yet, the F1 and LFS metrics utilized in this paper rely entirely on matching against ground-truth references.
>
> **A2**: We thank the reviewer for raising this critical point. We agree that this is exactly one of the central challenges in evaluating open-ended question answering, and in fact, it is a key issue that our work aims to address.
>
> Open-ended questions do not have a unique, universally accepted standard answer. Therefore, our goal is **not** to impose a single “correct answer” on such questions. Instead, we shift the focus of evaluation from **matching one standard answer** to **measuring whether the model sufficiently covers the relevant information within a controlled setting**.
>
> Although open-ended questions do not admit a unique answer, in our specifically constructed **local search environment**, the amount of topic-relevant information is **finite and discoverable**. Based on this property, we do not construct a single ground-truth answer. Instead, we build a **reference information pool** for evaluation. To make this pool as comprehensive as possible, we use multiple advanced LLMs to independently search for and synthesize information about the same topic, and then aggregate their outputs into a unified reference set.
>
> Regarding the evaluation metrics, our **F1** is also **not based on surface-level lexical overlap**. Instead, it is computed based on **semantic matching**. More specifically, we use sentence embeddings or “LLM as judge” to determine whether the information in the model output is supported by the reference information pool. This design reduces the influence of wording differences and stylistic variation, and makes the evaluation focus more on whether the model has truly retrieved and organized substantial information, rather than whether it reproduces a fixed piece of text.
>
> Therefore, our method does not reduce open-ended QA to simple “standard answer matching.” Rather, under the premise that open-ended questions lack a unique answer, we propose a **reference-guided information coverage evaluation** that makes the task operational and quantifiable.

---

> > ### Author Response · Authors · 2026-04-01
> > **Response to the reviewer: part2**
> >
> > **Q3**: Baseline models underwent no specific fine-tuning to adapt to the O²-QA format (specifically, the list of "key findings"). This disparity in methodology may result in an unfair comparison.
> >
> > **A3**: We thank the reviewer for raising this concern. We would like to clarify that our intention was not to amplify the advantage of our method through differences in output format. For the general-purpose LLM baselines, we adopted a **prompt-based setup**, in which the models were explicitly instructed to generate outputs in the target format before evaluation. We have also provided the **detailed system prompts used for these prompt-engineered LLMs in the appendix** of the original submission. Our primary goal was to assess their practical capability in **open-ended information retrieval, integration, and organization** under realistic usage scenarios. For API-based models in particular, this is also the most common and practically relevant evaluation setting.
> >
> > **Q4**: With a test set consisting of only 60 samples, it is statistically difficult to draw reliable conclusions.
> >
> > **A4**: We thank the reviewer for this important comment regarding dataset scale. To further examine the robustness and generalization ability of our method, we additionally constructed 75 manually verified, high-quality test questions for supplementary evaluation.
> >
> > Following the reviewer’s suggestion, we adopted a reference-free “LLM-as-Judge” evaluation for this additional test set. Specifically, for each question, we provided the question together with the outputs (obtained by searching the open web) from different models to an open-source judge model, **Qwen3.5-35B-A3B**, which was asked to score each answer on a scale of 1–10 and produce a ranking according to the following criteria:
> >
> > - **Relevance**: How well do the findings address the question?
> > - **Completeness**: Do the findings cover the key aspects of the topic?
> > - **Accuracy**: Are the findings factually correct and well grounded?
> > - **Depth & Insight**: Do the findings go beyond surface-level observations?
> > - **Clarity**: Are the findings clearly and concisely expressed?
> >
> > We evaluated DeepSeek-v3.2, Gemini-2.5-flash, GPT-4o-mini, Qwen2.5-72B-Instruct, Doubao-1.5-pro-32k, and our O2-Searcher on this held-out subset. The results are summarized in the table below.
> >
> > | Model | Avg. Score | Avg. Ranking |
> > |-------|-----------|-------------|
> > | DeepSeek-v3.2 | 8.88 | 1.97 |
> > | Gemini-2.5-flash | 8.20 | 3.20 |
> > | O2-Searcher | 7.99 | 3.57 |
> > | GPT-4o-mini | 7.73 | 3.81 |
> > | Qwen2.5-72B | 7.64 | 4.13 |
> > | Doubao-32K | 7.43 | 4.31 |
> >
> > These results provide an additional perspective on generalization. While O2-Searcher shows an advantage over models such as Qwen2.5-72B and GPT-4o-mini, it still trails behind stronger and more recent frontier models such as DeepSeek-v3.2. We explain this is because, for open-ended QA, extensive pretraining knowledge and larger model scale still play an important role in determining answer quality.
> >
> > We have clarified these points in the revision.
> >
> > **Q5**: typo erros.
> >
> > **A5**: Thank you for your careful reading. We have proofread the paper and corrected the corresponding typos.

---

### Review · Reviewer_DjrU · 2026-03-23

**Summary Of Contributions:**

This paper introduces $O^2$-Searcher, a reinforcement learning-based search agent for both open-ended and closed-ended question answering. The key contributions are a locally simulated search environment that reduces training cost, a composite reward function tailored to open-ended QA, and $O^2$-QA, a manually curated 300-question benchmark with ~30k cached web pages. A 3B-parameter model outperforms larger prompted LLM baselines on  $O^2$-QA and matches 7B-scale models on closed-ended benchmarks. Strengths include clear motivation and a thoughtful reward design. Weaknesses include evaluation on only one model size and a circular relationship between the training corpus and the benchmark.

**Audience:**

Yes

**Audience Explanation:**

Open-ended QA is underexplored relative to closed-ended work, and the combination of RL training, a local search environment, and a new benchmark addresses a genuine gap. Researchers in LLM agents, RAG, and generative evaluation methodology will find the contributions relevant.

**Broader Impact Concerns:**

No Broader Impact statement is included. Given that the system autonomously retrieves and synthesizes web content, a brief discussion of misinformation risks and responsible deployment would be appropriate.

**Claims And Evidence:**

Yes

**Claims Explanation:**

Tables 1 and 2 offer broad comparisons and Table 3 ablates reward components meaningfully. However, $O^2$-QA is constructed from the same cached corpus and the same prompt-engineered pipeline used for training, which risks measuring in-distribution performance rather than genuine open-ended generalization. Additionally, all experiments use a single 3B backbone, making it impossible to assess whether gains generalize across scales. The condensation module, described as central to the pipeline, is never ablated in isolation.

**Requested Changes:**

The most critical issue is benchmark validity. Since $O^2$-QA questions, ground-truth answers, and training data all derive from the same pipeline, the authors should provide evidence that test-time retrievals are not in-distribution with training content, or include a held-out subset with independently sourced answers.

A second critical concern is the single-scale evaluation. Even one experiment with a 7B backbone would clarify whether the observed advantages stem from the training framework or from properties specific to the 3B model.

---

> ### Author Response · Authors · 2026-04-01
> **Response to the reviewer: part1**
>
> Dear Reviewer DjrU:
>
> Thank you for your thorough review. We are grateful for your highlighting the clear motivation and contributions of our methodology. We provide discussions and explanations about your concerns as follows.
>
> **Q1**: Provide evidence that test-time retrievals are not in-distribution with training content, or include a held-out subset with independently sourced answers.
>
> **A1**: We thank the reviewer for this important suggestion. Our original intention is that, for open-ended question answering, the goal is not merely to match a single standard answer, but rather to evaluate a model’s ability to retrieve, synthesize, and organize relevant information. For this reason, instead of relying on a single ground-truth answer, we adopt a reference information pool as the basis for evaluation. This pool is constructed by aggregating outputs from multiple different LLMs, each of which independently searches for and synthesizes information under the same topic. The purpose is to improve coverage and reduce the bias introduced by any single generator.
>
> At present, our training and validation data are drawn from diverse sources spanning multiple domains. Moreover, at inference time, we observe consistent performance trends across different retrieval environments, including both local corpora and the open web. This provides some evidence that our method does not depend on a single retrieval setting.
>
> In addition, to further examine generalization, we constructed a manually verified set of 75 high-quality test questions. Following the suggestion from reviewer juij, we conducted a reference-free “LLM-as-Judge” evaluation to assess and rank the outputs produced by different models. Specifically, we provided the test questions together with the corresponding model outputs (obtained by searching the open web) to the judge model **Qwen3.5-35B-A3B**, which was instructed to score each response on a scale of 1–10 and rank them according to the following criteria:
>
> - **Relevance**: How well do the findings address the question?
> - **Completeness**: Do the findings cover the key aspects of the topic?
> - **Accuracy**: Are the findings factually correct and well grounded?
> - **Depth & Insight**: Do the findings go beyond surface-level observations?
> - **Clarity**: Are the findings clearly and concisely expressed?
>
> We evaluated DeepSeek-v3.2, Gemini-2.5-flash, GPT-4o-mini, Qwen2.5-72B-Instruct, Doubao-1.5-pro-32k, and our O2-Searcher on this held-out subset. The results are summarized in the table below. Although our model shows some advantage over models such as Qwen2.5-72B and GPT-4o-mini, it still lags behind more recent models such as DeepSeek-v3.2 in terms of generalization. We explain this is because, for open-ended QA, extensive pretraining knowledge and larger model scale still play an important role in determining answer quality.
>
> We have clarified these points in the revision.
>
> | Model              | Avg. Score | Avg. Ranking |
> |-------------------|----------:|------------:|
> | DeepSeek-v3.2     | 8.88      | 1.97        |
> | Gemini-2.5-Flash  | 8.20      | 3.20        |
> | O2-Searcher       | 7.99      | 3.57        |
> | GPT-4o-mini       | 7.73      | 3.81        |
> | Qwen2.5-72B       | 7.64      | 4.13        |
> | Doubao-32k        | 7.43      | 4.31        |

---

> ### Author Response · Authors · 2026-04-01
> **Response to the reviewer: part2**
>
> **Q2**: Experiment with a 7B backbone would clarify whether the observed advantages stem from the training framework or from properties specific to the 3B model.
>
> **A2**: We thank the reviewer for this valuable suggestion. First, we would like to reiterate the model performance from our initial submission. Through our innovative training approach, which combines open-ended and closed-ended questions, our 3B model achieves superior performance on multiple challenging QA benchmarks. And we have already provided results with a **7B backbone** on **closed-ended QA** in the original submission. We apologize for the oversight of the corresponding open-ended QA results.
>
> For clarity, we provide the scaling results on both types of questions below.
>
> Table 1: Performance on Closed-Ended QA Benchmarks
>
> | Base Model              | NQ    | HotpotQA | TriviaQA | PopQA | 2wiki | Musique | Bamboogle | Avg.   |
> |-------------------------|-------|----------|----------|-------|-------|----------|-----------|-------|
> | Qwen2.5-3B-Instruct     | 0.444 | 0.388    | 0.597    | 0.429 | 0.374 | 0 0.160 | 0.344     | 0.391 |
> | Qwen2.5-7B-Instruct    | 0.475 | 0.423    | 0.641    | 0.458 | 0.400 | 0.191    | 0.440     | 0.433 |
>
> Table 2: Performance on our Open-Ended O2-QA Benchmark
>
> | Base Model              | Easy  | Hard  | Avg.    |
> |-------------------------|-------|-------|--------|
> | Qwen2.5-3B-Instruct    | 0.2696| 0.1743| 0.2236 |
> | Qwen2.5-7B-Instruct    | 0.2946| 0.2149| 0.2561 |
>
> As shown in the tables, scaling from 3B to 7B parameters significantly improves performance: 10.7% on the closed benchmark (0.391 → 0.433) and 14.5% on our O2-QA benchmark (0.2236 → 0.2561). This demonstrates the potential scalability of our approach, with larger models benefiting more from our training method on complex open-ended queries.
>
>
> **Q3**: Ablation on the condensation module
>
> **A3**: We thank the reviewer for this valuable suggestion. In the revision, we have added an ablation study to isolate the effect of the condensation module.
>
> In our original setup, we use Qwen2.5-72B-Instruct as the default condensation module. To better understand the impact of this design choice, we replace it with several alternative models and compare their performance. The results show that using different condensation models can affect downstream QA performance, and some stronger models achieve better results than the default Qwen2.5-72B-Instruct setting.
>
> In addition, we also examine whether increasing the amount of information produced by the condensation module is beneficial. Specifically, we double the amount of structured learnings in its output relative to the original setting. As shown below, generating more condensed learnings further improves performance, suggesting that the quality and quantity of the condensed intermediate output both play important roles in the final results.
>
> | Condensation Module                         | Easy   | Hard   | Avg.    |
> |--------------------------------------------|--------|--------|--------|
> | Qwen2.5-72B-Instruct (More learnings)     | 0.3216 | 0.1869 | 0.2565 |
> | Qwen2.5-72B-Instruct                       | 0.2696 | 0.1743 | 0.2236 |
> | GPT-4o-mini                                 | 0.2868 | 0.1872 | 0.2386 |
> | Gemini-2.5-flash                            | 0.2699 | 0.1900 | 0.2313 |
> | Deepseek-v3                                 | 0.2826 | 0.1909 | 0.2383 |

---

### Decision · Action_Editor_rW56 · 2026-05-13

**Recommendation:** Accept with minor revision

**Additional Comments:**

The paper makes an interesting contribution to open-domain and open-ended QA with the proposed framework and benchmark. They addressed most of the reviewer's comments.
I therefore recommend minor revision, provided that the authors add a discussion on the hyperparameters of the reward function. I understand that in the current implementation, they are selected through empirical tuning. But I believe that the authors could discuss the hyperparameters in the appendix, to understand the sensitivity to these choices and the impact of the reward components and highlight that the proposed choice can be generalized.

**Audience:**

Yes

**Audience Explanation:**

Open-ended QA is comparatively underexplored relative to closed-ended question answering.
In addition, the combination of reinforcement learning with a local search environment, and the introduction of a dedicated benchmark addresses an interesting problem for a large audience.

**Claims And Evidence:**

Yes

**Claims Explanation:**

The authors introduce a search agent for open-domain question answering, and set the focus on challenging open-ended problems. They propose a local search environment that simulates an online search engine in response to agent-generated queries. They design a reward function to guide the agent toward solving both open-ended and closed-ended questions. The main contribution of this paper is the unified framework that enables a single agent to handle different types of questions efficiently. In addition, the authors propose a QA benchmark specifically designed to assess the ability of large language models to answer complex open-ended questions.

During the rebuttal period, reviewers were concerned with several weaknesses that should be addressed. In particular, concerning the proposed methodology, they asked for stronger evidence (test-time samples not in-distribution with respect to the training set, clarifications on the baselines on closed-ended and open-ended benchmarks, ablation study). They also found that the evaluation is limited, with a test set consisting of only 60 samples which raises questions about scalability and reproducibility.  They also asked for clarifications about the overall framework and evaluation protocol (answer matching) and hyperparameter tuning.

The authors mostly addressed these concerns with an additional set of 75 high-quality test questions, and provided additional comments on the performance of the approach with respect to various models. They also provided additional evaluation to strengthen the empirical evidence and alleviate concerns about the evaluation protocol.